# Lifelong Embodied Navigation Learning

**Xudong Wang**[*,1,2], **Jiahua Dong**[*,3], **Baichen Liu**[†,1], **Qi Lyu**[1,2], **Lianqing Liu**[1], **Zhi Han**[†,1],

[1] State Key Laboratory of Robotics and Intelligent Systems, Shenyang Institute of Automation,
  Chinese Academy of Sciences,
[2] University of Chinese Academy of Sciences,
[3] Mohamed bin Zayed University of Artificial Intelligence

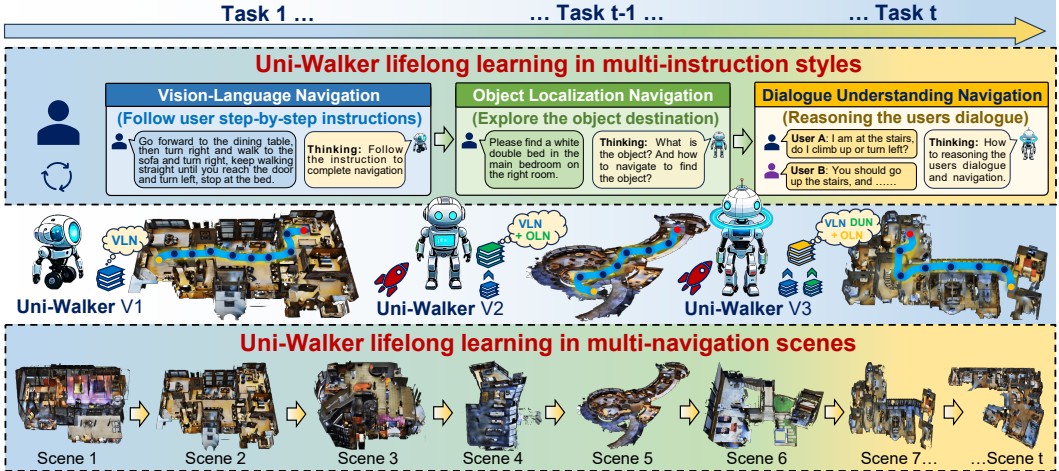

Figure 1: Illustration of the proposed lifelong embodied navigation learning (LENL) task. The proposed Uni-Walker is able to evolve and continually learn multiple new navigation tasks based on learned navigation knowledge, for developing universal embodied navigation. In lifelong learning, the sequential navigation tasks include multi-scenes and multi-instruction styles (VLN, OLN, DUN).

## Abstract

Embodied navigation agents powered by large language models have shown strong performance on individual tasks but struggle to continually acquire new navigation skills, which suffer from catastrophic forgetting. We formalize this challenge as lifelong embodied navigation learning (LENL), where an agent is required to adapt to a sequence of navigation tasks spanning multiple scenes and diverse user instruction styles, while retaining previously learned knowledge. To tackle this problem, we propose Uni-Walker, a lifelong embodied navigation framework that decouples navigation knowledge into task-shared and task-specific components with Decoder Extension LoRA (DE-LoRA). To learn the shared knowledge, we design a knowledge inheritance strategy and an experts co-activation strategy to facilitate shared knowledge transfer and refinement across multiple navigation tasks. To learn the specific knowledge, we propose an expert subspace orthogonality constraint together and a navigation-specific chain-of-thought reasoning mechanism to capture specific knowledge and enhance instruction-style understanding. Extensive experiments demonstrate the superiority of Uni-Walker for building universal navigation agents with lifelong learning. The code is available at **https://github.com/WangXudongSIA/Uni-Walker**.

## 1 Introduction

Embodied navigation has emerged as a fundamental problem in robotics Liang et al. (2025); Dong et al. (2026); Wang et al. (2026b), aiming to build agents that can follow user natural language instructions to reach destinations in visually complex scenes Liu et al. (2025). Beyond single-task

---

[*] Equal contribution. [†] Corresponding authors.

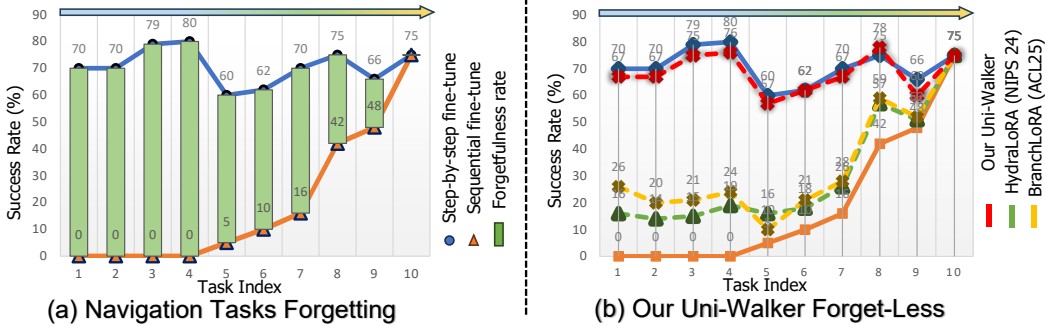

Figure 2: Illustration of the LENL performance. (a) The catastrophic forgetting phenomenon under the LENL settings. (b) Our proposed Uni-Walker has a better anti-forgetting performance.

navigation, universal embodied navigation aspires to develop a universal agent capable of understanding diverse instruction styles and adapting to a wide range of navigation scenes Gao et al. (2025b); Zheng et al. (2025a). Such a universal agent must flexibly solve different navigation tasks, including Vision-and-Language Navigation (VLN) Anderson et al. (2018a), Object-and-Language Navigation (OLN) Qi et al. (2020a), and Dialog Understanding Navigation (DUN) Thomason et al. (2020), demonstrating robust generalization across diverse scenes and instructions.

To develop a universal embodied navigation agent, recent methods leverage pretrained large language models (LLMs) Naveed et al. (2025) as the backbone and jointly fine-tune them on multiple navigation datasets. For example, NaviLLM Zheng et al. (2024), NavA3 Zhang et al. (2025b), SAME Zhou et al. (2024), and OctoNav Gao et al. (2025a) use LLMs to construct multi-task embodied navigation agents. Although these agents undergo large-scale joint training across multiple navigation tasks, they struggle to generalize across all tasks and often lack continual adaptability to ever-changing new navigation scenarios. In contrast, humans continually learn and absorb new knowledge through lifelong learning as they grow, allowing themselves to accumulate previously learned experiences over time and leverage these experiences to consecutively acquire new skills Zheng et al. (2025b). Inspired by this observation, we aim for navigation agents to continually evolve, learning to master diverse scenes and instruction styles for universal navigation, much like humans. When continually learning new navigation tasks, existing multi-task agents may suffer from catastrophic forgetting (see Fig. 2(a)) of previously learned tasks if there is insufficient memory and computational resources to store the full training data of old tasks and retrain the pretrained LLMs.

To mitigate catastrophic forgetting, a straightforward strategy is to employ Low-Rank Adaptation (LoRA) Hu et al. (2022) to fine-tune pretrained LLMs on each navigation task and store their task-specific low-rank weights for inference. Unfortunately, such a naive strategy fails to explore task-shared knowledge between new and old tasks, nor can it leverage the learning experiences accumulated from previous navigation tasks to enhance adaptation to new navigation tasks. Additionally, to capture the task-shared information across different navigation tasks, some recent studies, such as MoE-LoRA Gao et al. (2024); Chen et al. (2024); Dou et al. (2024), HydraLoRA Tian et al. (2024), and BranchLoRA Zhang et al. (2025a), propose multi-expert structures for jointly learning all tasks. However, they assume a fixed number of experts, which limits their scalability for lifelong learning and deployment in diverse real-world scenarios. To address the above challenges, we introduce Lifelong Embodied Navigation Learning (LENL) in this work, a novel problem where agents are required to continually adapt to a sequence of navigation tasks while retaining previously acquired navigation knowledge, ultimately developing into a universal embodied navigation agent (i.e., capable VLN, OLN, DUN), as Fig. 1. Inspired by the human lifelong learning process, which summarizes and exploits acquired knowledge to continually learn new information, we argue that the crucial challenges of efficient lifelong embodied navigation are consecutively learning new navigation tasks while mitigating the catastrophic forgetting of old ones under the LENL setting.

To resolve the challenges in the LENL problem, we build a new lifelong embodied navigation benchmark and propose a novel lifelong embodied navigation model named Uni-Walker. To the best of our knowledge, this paper is a pioneer exploration for lifelong embodied navigation learning. The proposed Uni-Walker model explicitly decouples multi-navigation task knowledge into shared and specific components with a new Decoder Extension LoRA (DE-LoRA). Inheriting and exploiting the shared knowledge, It learns new tasks more effectively during continual learning. To learn the shared navigation knowledge, we design a knowledge inheritance Strategy (KIS) and an experts co-activation strategy (ECAS) to facilitate task-shared knowledge transfer and refinement across

multiple navigation tasks. To learn the specific navigation knowledge, we propose an expert subspace orthogonality constraint (ESOC) and a navigation-specific chain-of-thought (NSCoT) reasoning mechanism to facilitate the learning of task-specific knowledge representations and tailor reasoning processes according to different instruction styles (VLN, OLN, DUN). Extensive experiments are performed to demonstrate that Uni-Walker can learn new navigation tasks without forgetting previously learned tasks, and even produce satisfactory generalization performance in unseen tasks, thus achieving the SOTA performance in LENL. The main contributions are outlined as follows:

• We introduce a novel Lifelong Embodied Navigation Learning (LENL) problem that enables navigation agents to continually learn new navigation tasks, including new scenes and instruction styles. We also build a new lifelong embodied navigation benchmark for training and evaluating.

• We propose Uni-Walker with a Decoder Extension LoRA (DE-LoRA) to decouple task-shared knowledge and task-specific knowledge across multiple navigation tasks (with multiple scenes and diverse instruction styles), to achieve efficient lifelong embodied navigation learning.

• We propose a Knowledge Inheritance Strategy (KIS) and an Experts Co-Activation Strategy (ECAS) to facilitate task-shared knowledge transfer and continual refinement across multiple navigation tasks, for exploring and exploiting the task-shared knowledge across multiple tasks.

• We propose an Expert Subspace Orthogonality Constraint (ESOC) to constrain each expert subspace to fully learn task-specific knowledge, and a Navigation Specific Chain of Thought (NSCoT) to provide specific chain of thought reasoning processes for each instruction style navigation, for exploring and exploiting the task-specific knowledge across multiple navigation tasks.

## 2 PROBLEM FORMULATION

**Preliminary:** In embodied navigation tasks, an agent operating in a 3D scene $\mathcal{S}$ is required to complete various tasks $\mathcal{T}$ described in user natural language instructions $\mathcal{I}$. Following recent researches Zheng et al. (2024); Zhou et al. (2024); Wei et al. (2025), there are three typical types of embodied navigation tasks: i). **Vision-Language Navigation (VLN)** Anderson et al. (2018a), denoted as $\mathcal{T}_{VLN}$, requires the agent to follow the user step-by-step instructions and navigate to the destination within the scene. ii). **Object Localization Navigation (OLN)** Qi et al. (2020a), denoted as $\mathcal{T}_{OLN}$, requires the agent to localize a distant destination object according to user concise high-level instructions. iii). **Dialogue Understanding Navigation (DUN)** Thomason et al. (2020), denoted as $\mathcal{T}_{DUN}$, requires the agent to understand the history dialogue from users and understand user requirements from it, and navigate to the destination. To construct a universal navigation agent for accomplishing multiple navigation tasks, following NavLLM Zheng et al. (2024), we use a pre-trained LLM, i.e., Vicuna Chiang et al. (2023), as the basic navigation agent $\mathcal{F}$. And for processing the agent's vision observation $\mathcal{O}$ during navigation, we use the CLIP Vision Transformer Encoder Sun et al. (2023) to extract vision features $\mathcal{V}(\mathcal{O})$, and the last hidden state embeddings of its [CLS] token are served as the agent real-time observation features. The architecture of our navigation agent model is similar to the multimodal large language model LLaVA Liu et al. (2023). At each time $i$, the navigation agent reasons user instruction $\mathcal{I}^i$ with the current observation $\mathcal{O}^i$ to output the the next direction: $D^i = \mathcal{F}(\mathcal{V}(\mathcal{O}^i), \mathcal{I}^i)$, thus navigation step by step to the destination. However, as shown in Fig. 1 (b), the agent's adaptation to a new navigation task $\mathcal{T}_t$ including new navigation scenes $\mathcal{S}_t$ or new user instruction styles $\{\mathcal{T}_{VLN,t}, \mathcal{T}_{OLN,t}, \mathcal{T}_{DUN,t}\}$ causes catastrophic forgetting of the old task $\{\mathcal{T}_1, \mathcal{T}_2, ..., \mathcal{T}_{t-1}\}$. This phenomenon limits the flexible deployment of embodied navigation agents.

**Problem Definition:** To address the above continual navigation learning challenges, we introduce a new problem setting, Lifelong Embodied Navigation Learning (LENL). We define multiple sequential navigation tasks as $\mathcal{T} = \{\mathcal{T}_1, \mathcal{T}_2, ..., \mathcal{T}_t\}$, where $t$-th navigation task $\mathcal{T}_t = \{\mathcal{O}_t, \mathcal{I}_t\}$ comprising multiple agent navigation vision observation $\mathcal{O}_t = \{o_1, o_2, ..., o_N\}$ in a specific navigation scene $\mathcal{S}_t \in \{S_1, S_2, ..., S_M\}$, and a specific user instruction style $\mathcal{I}_t \in \{\mathcal{T}_{VLN,t}, \mathcal{T}_{OLN,t}, \mathcal{T}_{DUN,t}\}$. The navigation agent $\mathcal{F}$ is required to learn all the navigation tasks $\mathcal{T}$ consecutively, and all the tasks are tested after all task learning is complete. Please note that for more practical applications of the LENL settings, the task-id $t$ is agnostic during the testing phase. And $\mathcal{S}_t$ of each $\mathcal{T}_t$ does not overlap with any previous tasks: $\mathcal{S}_t \bigcap (\bigcup_{j=1}^{t-1} \mathcal{S}_j) = \emptyset$. In summary, the proposed LENL problem aims to let a navigation agent $\mathcal{F}$ continually learn a sequence of new navigation tasks $\mathcal{T}$ while alleviating the forgetting of old tasks, thus facilitating the development of a universal embodied navigation agent.

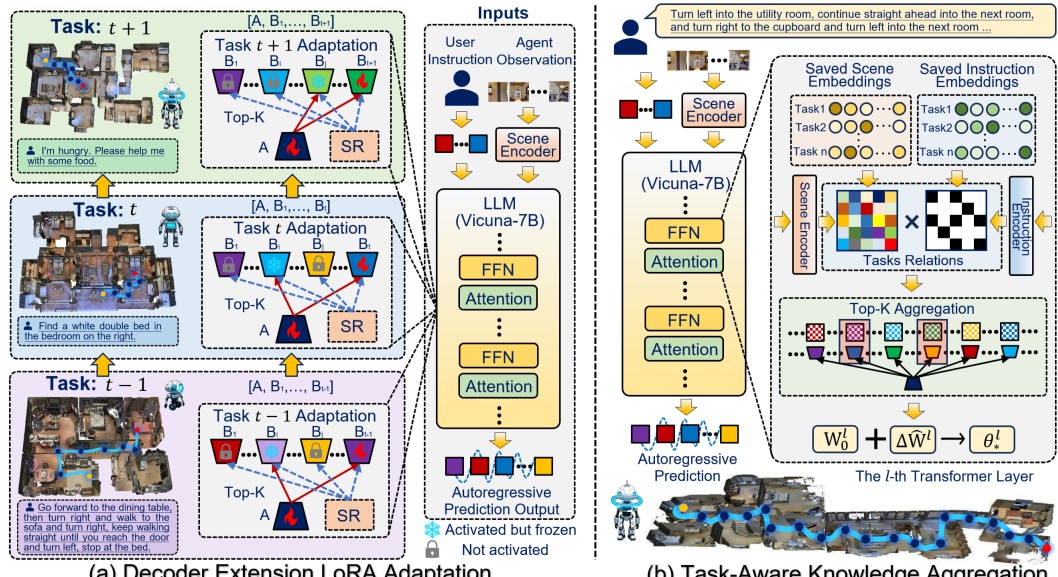

Figure 3: Illustration of the proposed Uni-Walker pipeline. It includes (a) a *Decoder Extension LoRA Adaptation* to achieve progressive knowledge decoupled learning, which decouples navigation knowledge into shared and specific parts, thereby facilitating new tasks learning using shared knowledge while avoiding forgetting. (b) a *Task-Aware Knowledge Aggregation* to automatically aggregate the learned knowledge according to a specific navigation task for task specific inference.

## 3   THE PROPOSED UNI-WALKER

The overall pipeline of our Uni-Walker is illustrated in Fig. 4. It includes a Decoder Extension LoRA (DE-LoRA) adaptation to learn a series of multiple navigation tasks continually (§3.1), a Navigation Chain-of-Thought (NCoT) to facilitate specific navigation reasoning (§3.2), and a Task-Aware Knowledge Aggregation (TAKA) to automatically aggregate the learned knowledge (§3.3).

### 3.1   DECODER EXTENSION LORA ARCHITECTURE

As shown in Fig. 4 (a), in order to learn the $t$-th navigation task $\mathcal{T}_t$, we extend the vanilla Lora Hu et al. (2022) and propose a new low-rank adapter named Decoder Extension LoRA (DE-LoRA) to finetune the pretrained NavLLM $\mathcal{F}_{\theta_0}$ on each navigation task $\mathcal{T}_t = \{\mathcal{O}_t, \mathcal{I}_t\}$, and then obtain an adapted updated model $\mathcal{F}'_{\theta'_t}$, where $\theta'_t = \theta_0 + \Delta\theta_t$, $\Delta\theta_t = \{\Delta\mathbf{W}^l_t\}^L_{l=1}$, and $\Delta\mathbf{W}^l_t = \mathbf{B}^l_t\mathbf{A}^l_t \in \mathbb{R}^{b_l \times a_l}$ are learnable low-rank weight in $l$-th layer of a total of $L$ layers. In this case, the features $x^l$ of each transformer layer are processed as: $y^l = \mathbf{W}^l_0 x^l + \mathbf{B}^l_t \mathbf{A}^l_t x^l$, where the low-rank learnable weights $\mathbf{B}^l_t \in \mathbb{R}^{b_l \times r}$ can be regarded as decoder and $\mathbf{A}^l_t \in \mathbb{R}^{r \times a_l}$ can be regarded as encoder. To address LENL, a trivial solution is to train and save all low-rank adaptation weights for all navigation tasks so far, and then load the specific weights at inference time. However, multiple navigation tasks share and have specific knowledge with each other, and simple vanilla LoRA storing for a single task cannot enable the agent to evolve during learning, limiting the development of a universal navigation agent. To explore and exploit the task-shared and task-specific knowledge between navigation tasks, different from vanilla LoRA, we use subspace $\mathbf{A}$ to learn task-shared knowledge and $\mathbf{B}_t$ to learn task-specific knowledge, inspired by Zhang et al. (2025a); Tian et al. (2024). Specifically, in our DE-LoRA, all navigation tasks use a shared subspace $\mathbf{A}$ and each navigation task uses TOP-K activated expert subspace $\{\mathbf{B}_1, ..., \mathbf{B}_K\}$ to exploit task-shared knowledge and to explore task-specific knowledge:

$$ y = \mathbf{W}_0 \cdot x + \Delta\mathbf{W} \cdot x = \mathbf{W}_0 \cdot x + \sum_{n=1}^{K}(\mathbf{B}_{t,n} \cdot \mathbf{A} \cdot x), \tag{1}$$

where $\mathbf{B}_n \in \{\mathbf{B}_1, ..., \mathbf{B}_K\}$ is the specific activated expert subspace. Different from existing shared architecture HydraLoRA Tian et al. (2024), our DE-LoRA selects sparse activated experts to adapt to the new navigation task and expands a specific expert subspace as the task is continually learned.

### 3.2   EXTENSION EXPERTS INCREMENTAL LEARNING

To explore the task-shared and task-specific knowledge between sequential navigation tasks set $\mathcal{T}$, and consolidate the DE-LoRA architecture, we propose an Extension Experts Incremental Learning

strategy. Specifically, for the initial training, we use Kaiming initialization He et al. (2015) to initialize knowledge base parameters $\mathcal{K} = \{\mathbf{A}, \mathbf{B}_1\}$, and then train the parameters $\{\mathbf{A}, \mathbf{B}_1\}$ to adapt task $\mathcal{T}_1$. And then for learning the subsequent $n$-th task $T_n$, we dynamically increase one $\mathbf{B}_n$ to $\mathcal{K}$.

**Inheritance Task-Shared Knowledge.** When learning the subsequent new task $\mathcal{T}_t = \{S_e, \mathcal{I}_s\}, t > 1$ continually (with $e$-th navigation scene $S_e$ and $s$-th navigation instruction style $\mathcal{I}_s$), we propose *Knowledge Inheritance Strategy* (KIS) and *Experts Co-Activation Strategy* (ECAS) on the new increased expert subspace $\mathbf{B}_t$ to explore task-shared knowledge. In order to inherit user instruction style knowledge, in KIS, we initialize the current expert $\mathbf{B}_t$ using the learned experts having the same instruction knowledge, denoted as $\{\mathbf{B}_i, \mathbf{B}_{i+1}, ..., \mathbf{B}_j\}$, that have previously learned tasks with the same instruction style. We perform Principal Component Analysis (PCA) Abdi & Williams (2010) on these experts to identify the low-dimensional subspace that captures the most significant shared variations among them. Specifically, each expert parameter matrix $\mathbf{B}_k \in \{\mathbf{B}_i, ..., \mathbf{B}_j\}$ is flattened into a vector $\theta_k = \text{vec}(\mathbf{B}_k) \in \mathbb{R}^d$, and all vectors are concatenated to form an expert matrix $\mathbf{M} = [\theta_i, \theta_{i+1}, \ldots, \theta_j] \in \mathbb{R}^{d \times (j-i+1)}$. Then we compute the mean parameter vector $\boldsymbol{\mu} = \frac{1}{j-i+1} \sum_{k=i}^{j} \theta_k$, and obtain centered vectors $\tilde{\theta}_k = \theta_k - \boldsymbol{\mu}$. The covariance matrix is given by $\mathbf{C} = \frac{1}{j-i+1} \sum_{k=i}^{j} \tilde{\theta}_k \tilde{\theta}_k^\top$, then we perform SVD decomposition on the covariance matrix $\mathbf{C} = \mathbf{U}\boldsymbol{\Lambda}\mathbf{V}^\top$ and take the top-$r$ eigenvectors to form $\mathbf{U}_r = [u_1, ..., u_r] \in \mathbb{R}^{d \times r}$. We further leverage the top-$r$ principal components to capture the dominant shared variations across previous experts. Specifically, we compute the average direction of the subspace spanned by $\mathbf{U}_r$ and use it for knowledge inheritance:

$$\mathbf{B}_t \leftarrow \hat{\mathbf{B}} = \text{mat}_{b \times r}(\boldsymbol{\mu} + \frac{1}{r} \sum_{k=1}^{r} u_k), \tag{2}$$

where $\boldsymbol{\mu}$ provides the central tendency of expert parameters, while the averaged principal directions inject additional shared variation patterns. This initialization encourages the new expert to start not only from the common mean but also aligned with the principal subspace that encodes instruction-style knowledge. In addition to KIS, we also propose ECAS to explore shared knowledge between multiple navigation tasks. Specifically, as shown in Fig. 4 (a), we activate TOP-K experts with the TAKA strategy (§3.3) for each task, which includes not only the specific trainable $\mathbf{B}_t$ expert, but also the related experts $\{\mathbf{B}_1^*, ..., \mathbf{B}_{K-1}^*\}$ that are computed in the forward pass but parameter frozen:

$$y = \mathbf{W}_0 \cdot x + \Delta\mathbf{W} \cdot x = \mathbf{W}_0 \cdot x + \mathbf{B}_t \cdot \mathbf{A} \cdot x + \sum_{n=1}^{K-1} (\mathbf{B}_n^* \cdot \mathbf{A} \cdot x), \tag{3}$$

In addition to the exploration shared knowledge with TOP-K expert subspace $\{\mathbf{B}_1^*, ..., \mathbf{B}_{K-1}^*\}$ and $\mathbf{B}_t$, in order to progressively refine the shared subspace $\mathbf{A}'$ and avoid old task-specific knowledge catastrophic forgetting, we also propose a shared smoothing consolidation loss for the subspace $\mathbf{A}'$:

$$\mathcal{L}_{ssc,t} = \lambda_{ssc}(\|F_{\mathbf{A},t-1} \odot (\mathbf{A}' - \mathbf{A})\|_F^2), \tag{4}$$

where $\mathbf{A}'$ is current learnable shared subspace, $\mathbf{A}$ is the last task $\mathcal{T}_{t-1}$ learned subspace. And $\lambda_{ssc} = 0.1$ is the balance hyper-parameter, $F_{t-1}$ is the Fisher Information Matrix Kirkpatrick et al. (2017) which measures the importance of each parameter $\mathbf{A}$ for the last task $\mathcal{T}_{t-1}$, and it can be calculated:

$$F_{\mathbf{A},t-1} = \mathbb{E}_{(x,y) \sim T_{t-1}} \Big[ \big(\partial_{\mathbf{A}} \log p(y \mid x; \mathbf{A})\big)^2 \Big], \tag{5}$$

where $x \in \{\mathcal{O}_t, \mathcal{I}_t\}$ is the navigation input data and $y$ is the output annotation, and $F_{\mathbf{A},t-1}$ measures the importance of each parameter for the navigation agent $\mathcal{F}$ output performance, with a higher value indicating greater importance. Furthermore, during the lifelong navigation learning, we also perform incremental updates for the Fisher Matrix $F_t$ in to achieve shared knowledge smooth learning:

$$F_{\mathbf{A},t} = \omega \cdot F_{\mathbf{A},t-1} + (1 - \omega) \cdot F_{\mathbf{A},t}, \tag{6}$$

where $\omega = 0.9$ is the exponential moving average coefficient to control the smooth update of $F_{\mathbf{A},t}$.

**Exploration Task-Specific Knowledge.** The exploration and exploitation of the above-mentioned task-shared knowledge summarizes old task knowledge to facilitate new task learning while avoiding old knowledge catastrophic forgetting. To further explore the independence of specific knowledge, we also propose an Expert Subspace Orthogonality Constraint (ESOC) and a Navigation Specific Chain of Thought (NSCoT). Specifically, to consolidate the shared architecture of subspace $\mathbf{A}$ and facilitate task-specific knowledge learning, we propose ESOC to perform orthogonal constraint on expert subspace $\mathbf{B}_t$: during learning the $t$-th task, we prefer the expert subspace $\mathbf{B}_t$ be orthogonal

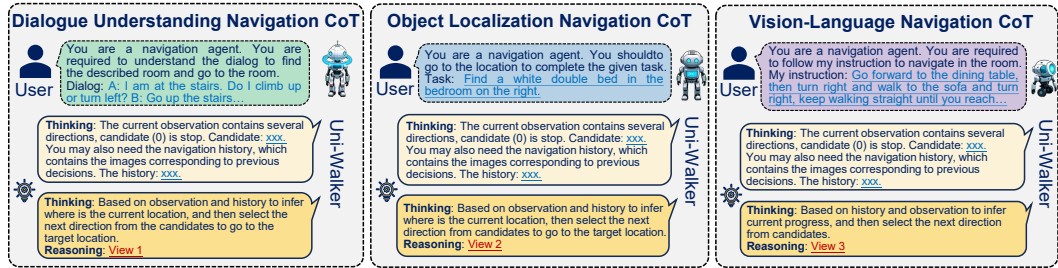

Figure 4: Illustration of the *Navigation Chain-of-Thought* to design various specific LLM chains of thought for specific instruction style tasks to facilitate the embodied navigation performance.

to the previous experts avoid knowledge overlap: $\sum_{i=1}^{t-1} || \operatorname{tr}((\mathbf{B}_i)^{\mathrm{T}}\mathbf{B}_t)||_F^2 = 0$. In addition, to avoid the expert subspace $\mathbf{B}_t$ degenerating into the trivial solution of a zero matrix (in this case, it will not learn any knowledge, i.e., learnable weight $\Delta\mathbf{W} = 0$), we perform L2 normalization on the expert subspace $\mathbf{B}_t$ to project these experts onto the unit sphere subspace $\widetilde{\mathbf{B}}_t$. Thus, the ESOC loss is:

$$\mathcal{L}_{esoc,t} = \lambda_{esoc} \sum_{i=1}^{t-1} |tr((\widetilde{\mathbf{B}}_i)^{\mathrm{T}}\widetilde{\mathbf{B}}_t)|, \quad \widetilde{\mathbf{B}}_i = \frac{\mathbf{B}_i}{\| \mathbf{B}_i \|_F + \epsilon}, \quad \widetilde{\mathbf{B}}_t = \frac{\mathbf{B}_t}{\| \mathbf{B}_t \|_F + \epsilon}. \quad (7)$$

where $\lambda_{esoc} = 0.1$ is the balance subspace orthogonality hyper-parameter, and $\epsilon = 0.01$ is the perturbation for avoiding the computational avalanches caused by molecules $\| \mathbf{B}_t \|_F^2$ equal to 0.

In addition to ESOC, we also propose NSCoT to perform specific navigation reasoning processes for each specific user instruction style navigation task. Specifically, as shown in Fig. 4 (b), we provide three types of chain of thought for three types of specific user instructions $\{\mathcal{T}_{VLN,t}, \mathcal{T}_{OLN,t}, \mathcal{T}_{DUN,t}\}$. In VLN task, the agent is required to follow the user's detailed step-by-step instructions, and the reasoning tends to be navigation process tracking. In OLN task, the agent is required to reason about the user's desired destination based on observations and gradually track the navigation process to reach the destination. In DUN task, the agent is required to comprehend user dialogues, reason user requirements, and complete navigation based on the requirements.

In summary, during the lifelong navigation learning process for the $t$-th task $\mathcal{T}_t$, the total adaptation loss for the LLM-based navigation agent $\mathcal{F}$ performing auto-regressive action generation training is:

$$\mathcal{L}_t = -\lambda \sum_{n=1}^{N} log P_t(A_n, \hat{\mathcal{P}}_{n|\mathcal{I},\mathcal{O}}) + \mathcal{L}_{ssc,t} + \mathcal{L}_{esoc,t}, \quad (8)$$

where $P_t(A_n, \hat{\mathcal{P}}_{n|\mathcal{I},\mathcal{O}})$ denotes the predicted probability of annotation navigation actions under the current observation $\mathcal{IO} = \{\mathcal{I}_t, \mathcal{O}_t\}$, and the balance hyper-parameter is $\lambda = 1 - (\lambda_{ssc} + \lambda_{ewc})$.

### 3.3 TASK-AWARE KNOWLEDGE AGGREGATION

After learning a series of navigation tasks continually, Uni-Walker is required to recall which expert knowledge learned is valuable for the current specific navigation task and utilize this knowledge to better complete the current navigation task. We propose a Task-Aware Knowledge Aggregation (TAKA) strategy to automatically activate the TOP-K experts most relevant to the current task. Specifically, as shown in Fig. 4 (c), Uni-Walker adapts to each task $\mathcal{T} = \{\mathcal{T}_1, \mathcal{T}_2, ..., \mathcal{T}_H\}$ by learning specific expert subspaces $\{\mathbf{A}, \{\mathbf{B}_1, \mathbf{B}_2, ..., \mathbf{B}_H\}\}$ while also preserving the task retrieval embedding $\mathcal{R}e = \{(E_{S,1}, E_{I,1}), (E_{S,2}, E_{I,2}), ..., (E_{S,H}, E_{I,H})\}$, where $E_{S,t}$ are the $t$-th task's scene embedding and $E_{I,t}$ are its instruction embedding. $E_{S,t} = EV(\mathcal{O}_t^1, \mathcal{O}_t^2, ..., \mathcal{O}_t^M)$ are the embedding observed by the agent during navigation, using the CLIP vision encoder Radford et al. (2021). $E_{I,t} = ET(\mathcal{I}_t^1, \mathcal{I}_t^2, ..., \mathcal{I}_t^N)$ are the embedding provided by user instructions in $t$-th task, using the CLIP text encoder. And to reduce storage space, we remove elements in $\{E_{S,t}, E_{I,t}\}$ with cosine similarity greater than 0.9. Under the LENL setting, the agent is agnostic about the task-id during inference, so we use $\mathcal{R}e$ to retrieve an unknown navigation task to decide which expert subspace to use. We compute the cosine similarity for each $E_{S,t}$ based on the agent's observation embeddings $E_o = EV(\mathcal{O}_q)$ and each $E_{I,t}$ based on the embedding provided by user instruction $E_i = ET(\mathcal{I}_q)$:

$$Sm_{O,t}(E_{S,t}, E_o) = \frac{E_o \cdot E_{S,t}}{\| E_o \|_F^2 \cdot \| E_{S,t} \|_F^2}, \; Sm_{I,t}(E_{I,t}, E_i) = \frac{E_i \cdot E_{I,t}}{\| E_i \|_F^2 \cdot \| E_{I,t} \|_F^2}, \; t = 1, 2, ..., H. \quad (9)$$

Subsequently, we set elements greater than $\mu$ in $\{Sm_{I,1}, Sm_{I,2}, ..., Sm_{I,H}\}$ to 1 and other elements to 0 to form a mask matrix $\mathcal{M}$. We apply mask $\mathcal{M}$ to similarity set $\{Sm_{O,1}, Sm_{O,2}, ..., Sm_{O,H}\}$,

then select the TOP-K experts $\{\mathbf{B}_q^*\}_{q=1}^k$ to activate to perform inference for the test agnostic task $T_q$:

$$\{\mathbf{B}_q^*\}_{q=1}^k = \text{top-}k(\mathcal{M} \odot \{Sm_{O,1}, Sm_{O,2}, ..., Sm_{O,H}\}), \tag{10}$$

Finally, activated experts $\{\mathbf{B}_q^*\}_{q=1}^k$ are combined with subspace $\mathbf{A}$ to perform inference with Eq.(1).

## 4 EXPERIMENTS

### 4.1 IMPLEMENTATION DETAILS

**Lifelong Embodied Navigation Benchmark Settings:** To evaluate the proposed LENL task, we follow the classic works on visual navigation Anderson et al. (2018a); Qi et al. (2020a); Thomason et al. (2020); Zheng et al. (2024) using the Matterport3D Simulator Anderson et al. (2018a) as navigation scene simulator, and construct a new lifelong navigation LENL benchmark. The proposed benchmark settings are shown in Fig. 5, including eighteen different navigation scenes with three embodied navigation instruction styles, i.e., VLN, OLN, DUN, as described in the Problem Definition (§ 2). The first fifteen tasks are used for continual learning, while the last three tasks are used for generalization testing in unseen scenes. Please note that in the

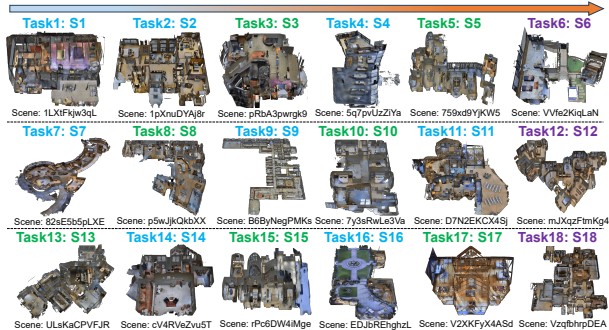

Figure 5: Illustration of the lifelong navigation benchmark. We establish a total of 18 navigation tasks for lifelong learning, including 18 navigation scenes and 3 types of user instruction styles (VLN is marked in blue color, OLN is green, and DUN is purple). We use the first 15 tasks for lifelong learning, and the last 3 tasks are used to evaluate unseen scene generalization performance.

LENL setting, the task-id $t$ is only visible during continual training, but agnostic during testing. More details for the constructed LENL benchmark are provided in our Appendix §C.

**Training and Evaluation Settings:** To ensure fair comparisons, our methods and all comparison methods are performed with NavLLM Zheng et al. (2024) as the baseline backbone. We use eight NVIDIA RTX 6000 Ada Generation GPUs with PyTorch 2.1.2 (cu121) for training and testing. And the Adam optimizer with an initial learning rate of $3.0 \times 10^{-5}$ is used for training. The low-rank $r = 16$ of LORA, the number of experts is $k = 2$, $\mu = 0.5$, and all the other hyperparameters are consistent with the NavLLM. Following previous studies Zheng et al. (2024); Wei et al. (2025); Anderson et al. (2018a), we report three key evaluation metrics: success rate (SR), success rate weighted by path length (SPL), and oracle success rate (OSR). In addition, to evaluate the anti-forgetting rate in lifelong learning, we propose three evaluation metrics, SR Forgetting Rate (SR-F), SPL Forgetting Rate (SPL-F), OSR Forgetting Rate (OSR-F). These metrics can be calculated by:

$$SR\text{-}F_t = \frac{M\text{-}SR_t - SR_t}{M\text{-}SR_t}, \ SPL\text{-}F_t = \frac{M\text{-}SPL_t - SPL_t}{M\text{-}SPL_t}, \ OSR\text{-}F_t = \frac{M\text{-}OSR_t - OSR_t}{M\text{-}OSR_t}, \tag{11}$$

where $M\text{-}SR_t$, $M\text{-}SPL_t$, and $M\text{-}OSR_t$ represents the $t$-th task completion status with success rate, success rate weighted by path length, and oracle success rate, under learning only the 1-st to $t$-th task. Thus the larger $SR\text{-}F_t$, $SPL\text{-}F_t$, and $OSR\text{-}F_t$ the greater the degree of $t$-th task forgetting.

### 4.2 COMPARISON EXPERIMENT RESULTS

This experiment verifies the superior performance of our Uni-Walker. We compare it against the SOTA LoRA-based continual learning approaches: **Seq-FT** is sequential fine-tuning across all tasks; **LwF-LoRA** Li & Hoiem (2017) employs knowledge distillation to preserve previously acquired tasks; **EWC-LoRA** Xiang et al. (2023) constrains important parameters of past tasks to reduce navigation forgetting; **Dense MoLE** Chen et al. (2024) adopts dense expert routing, while **Sparse MoLE** Dou et al. (2024) applies sparse expert routing in MoE-LoRA; **MoLA** Gao et al. (2024) extends Sparse MoLE by introducing deeper-level experts; **HydraLoRA** Tian et al. (2024) uses a shared module **A** for common knowledge and multiple **B** modules for task-specific learning, **BranchLoRA** Zhang et al. (2025a) further strengthens the sparse selection mechanism; **O-LoRA**

Table 1: Test results (task-wise **SR** ↑, %) of comparison experiment with our LENL settings.

| Comparison Methods | S1 | S2 | S3 | S4 | S5 | S6 | S7 | S8 | S9 | S10 | S11 | S12 | S13 | S14 | S15 | S16 | S17 | S18 | Avg. |
|---|---|---|---|---|---|---|---|---|---|---|---|---|---|---|---|---|---|---|---|
| Seq-FT (sequential fine-tuning) | 0 | 0 | 0 | 5 | 4 | 0 | 5 | 18 | 8 | 9 | 8 | 21 | 20 | 25 | 86 | 0 | 0 | 0 | 12 |
| LwF-LoRA (Li & Hoiem (2017)) | 0 | 5 | 0 | 5 | 6 | 0 | 5 | 22 | 8 | 12 | 10 | 30 | 45 | 36 | 85 | 5 | 0 | 0 | 15 |
| EWC-LoRA (Xiang et al. (2023)) | 0 | 5 | 0 | 10 | 4 | 0 | 5 | 20 | 8 | 9 | 8 | 31 | 49 | 40 | 86 | 5 | 5 | 0 | 16 |
| Dense MoLE (Chen et al. (2024)) | 10 | 10 | 4 | 10 | 8 | 3 | 10 | 18 | 16 | 19 | 17 | 30 | 56 | 51 | 85 | 12 | 10 | 8 | 21 |
| Sparse MoLE (Dou et al. (2024)) | 15 | 12 | 0 | 17 | 6 | 6 | 16 | 18 | 18 | 24 | 15 | 32 | 58 | 53 | 84 | 16 | 12 | 10 | 23 |
| MoLA (Gao et al. (2024)) | 15 | 12 | 0 | 12 | 6 | 10 | 20 | 18 | 13 | 26 | 19 | 34 | 61 | 59 | 86 | 20 | 12 | 11 | 24 |
| HydraLoRA (Tian et al. (2024)) | 16 | 14 | 15 | 19 | 6 | 15 | 16 | 24 | 17 | 17 | 21 | 41 | 68 | 57 | 86 | 18 | 14 | 16 | 27 |
| BranchLoRA (Zhang et al. (2025a)) | 26 | 20 | 20 | 25 | 8 | 21 | 20 | 26 | 20 | 18 | 22 | 39 | 78 | 54 | 86 | 28 | 20 | 15 | 30 |
| SEMA (Wang et al. (2025)) | 38 | 34 | 32 | 34 | 19 | 32 | 33 | 38 | 42 | 30 | 33 | 51 | 84 | 61 | 86 | 39 | 42 | 36 | 43 |
| NBAgent (Liang et al. (2024)) | 29 | 26 | 23 | 26 | 10 | 23 | 23 | 28 | 24 | 22 | 26 | 42 | 81 | 57 | 86 | 31 | 24 | 17 | 33 |
| O-LoRA (Wang et al. (2023a))+TAKA | 55 | 61 | 50 | 54 | 42 | 42 | 54 | 79 | 48 | 66 | 56 | **54** | 88 | 63 | 86 | 65 | 53 | 36 | 58 |
| SD-LoRA (Wu et al. (2025))+TAKA | 58 | 57 | 54 | 46 | 49 | 42 | 57 | 76 | 51 | 59 | 57 | **54** | 81 | 62 | **87** | 68 | 55 | 48 | 59 |
| **Uni-Walker (ours)** | **67** | **67** | **75** | **67** | **57** | **50** | **67** | **83** | **50** | **73** | **62** | 53 | **88** | **65** | 86 | **74** | **61** | **51** | **66** |

Table 2: Test results (task-wise **SR-F** ↓, %) of comparison experiment with our LENL settings.

| Comparison Methods | S1 | S2 | S3 | S4 | S5 | S6 | S7 | S8 | S9 | S10 | S11 | S12 | S13 | S14 | S15 | S16 | S17 | S18 | Avg. |
|---|---|---|---|---|---|---|---|---|---|---|---|---|---|---|---|---|---|---|---|
| Seq-FT (sequential fine-tuning) | 100 | 100 | 100 | 94 | 93 | 100 | 93 | 78 | 90 | 88 | 88 | 61 | 77 | 62 | 0 | 100 | 100 | 100 | 85 |
| LwF-LoRA (Li & Hoiem (2017)) | 100 | 93 | 100 | 94 | 90 | 100 | 93 | 74 | 90 | 84 | 84 | 44 | 49 | 45 | 0 | 94 | 100 | 100 | 80 |
| EWC-LoRA (Xiang et al. (2023)) | 100 | 93 | 100 | 87 | 93 | 100 | 93 | 76 | 90 | 88 | 88 | 43 | 44 | 38 | 0 | 94 | 92 | 100 | 79 |
| Dense MoLE (Chen et al. (2024)) | 86 | 85 | 95 | 87 | 87 | 95 | 86 | 79 | 81 | 74 | 73 | 44 | 36 | 22 | 0 | 86 | 84 | 85 | 71 |
| Sparse MoLE (Dou et al. (2024)) | 78 | 82 | 100 | 79 | 90 | 89 | 77 | 79 | 79 | 68 | 77 | 41 | 34 | 18 | 0 | 81 | 81 | 82 | 69 |
| MoLA (Gao et al. (2024)) | 78 | 82 | 100 | 85 | 90 | 82 | 71 | 79 | 85 | 65 | 70 | 37 | 31 | 9 | 0 | 76 | 82 | 80 | 67 |
| HydraLoRA (Tian et al. (2024)) | 77 | 79 | 81 | 76 | 90 | 73 | 77 | 72 | 67 | 77 | 68 | 27 | 24 | 14 | 0 | 79 | 79 | 71 | 63 |
| BranchLoRA (Zhang et al. (2025a)) | 63 | 71 | 75 | 69 | 87 | 62 | 71 | 69 | 62 | 76 | 66 | 30 | 13 | 17 | 0 | 67 | 71 | 73 | 58 |
| O-LoRA (Wang et al. (2023a))+TAKA | 21 | 13 | 37 | 33 | 30 | 24 | 23 | 7 | 8 | 12 | 14 | 4 | 2 | 3 | 0 | 23 | 22 | 35 | 17 |
| SD-LoRA (Wu et al. (2025))+TAKA | 17 | 19 | 32 | 43 | 21 | 24 | 19 | 11 | **2** | 21 | 12 | 4 | 10 | 5 | 0 | 19 | 19 | 13 | 16 |
| **Uni-Walker (ours)** | **4** | **4** | **5** | **16** | **5** | **4** | **4** | **2** | **4** | **3** | **5** | **4** | **2** | **0** | **0** | **12** | **10** | **7** | **5** |

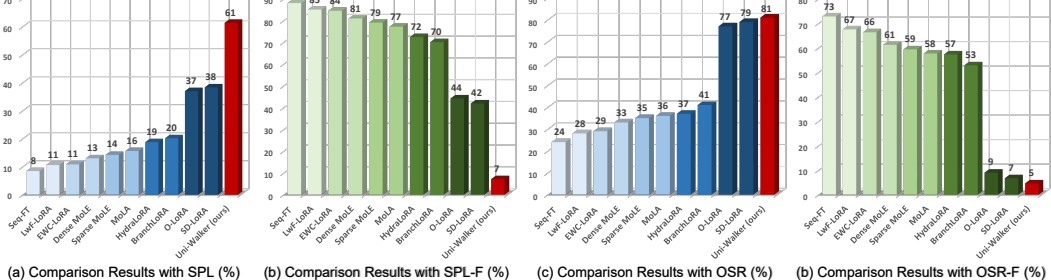

(a) Comparison Results with SPL (%)  (b) Comparison Results with SPL-F (%)  (c) Comparison Results with OSR (%)  (b) Comparison Results with OSR-F (%)

Figure 6: Test results (average (a) **SPL** ↑, (b) **SPL-F** ↓, (c) **OSR** ↑, (d) **OSR-F** ↓) of comparison experiment with our LENL settings. For detailed task-wise results, please refer to Appendix §C.

Wang et al. (2023a) leverages orthogonal loss to capture task-specific knowledge; **SD-LoRA** Wu et al. (2025) dynamically combines LoRA modules from previously learned skills. And SD-LoRA and O-LoRA are combined with our proposed TAKA. Additional comparison details are provided in the Supplementary Materials. The evaluation results with SR are summarized in Table 1 and Table 2. Uni-Walker achieves a higher average success rate of 66%, surpassing the previous best (59%) by 7%, and a lower forgetting rate of 5%, improving the prior best (16%) by 11%. The results with SPL and OS are summarized in Fig. 6. Uni-Walker achieves a higher SPL of 61%, surpassing the previous best (38%) by 23%, and a lower forgetting rate of 7%, improving the prior best (42%) by 35%. It achieves a higher OSR of 81%, surpassing the previous best (79%) by 2%, and a lower forgetting rate of 5%, improving the prior best (7%) by 2%. Detailed results are in Appendix §C.

**How Does Uni-Walker Perform on Generalization in Unseen Scenes?** We also provide a dedicated comparison on the three unseen generalization tasks (S16, S17, S18). The evaluation results with SR (%) are summarized in Table 3. Uni-Walker achieves a higher average success rate of 62%, surpassing the previous best (57%) by 5%. This superior generalization capability results from: Decoupled knowledge learning via DE-LoRA, ESOC, and KIS, ensuring that shared

Table 3: Results (SR%) for Generalization.

| Comparisons | S16 | S17 | S18 | Avg |
|---|---|---|---|---|
| HydraLoRA | 18 | 14 | 16 | 16.0 |
| BranchLoRA | 28 | 20 | 15 | 21.0 |
| O-LoRA + TAKA | 65 | 53 | 36 | 51.3 |
| SD-LoRA + TAKA | 68 | 55 | 48 | 57.0 |
| Uni-Walker (ours) | **74** | **61** | **51** | **62.0** |

Table 4: Ablation study results (%) for the methods with **Task Shared Knowledge Exploration**.

| Methods | KIS | ECAS | SSC | SR ↑ | SR-F ↓ | SPL ↑ | SPL-F ↓ | OSR ↑ | OSR-F ↓ |
|---|---|---|---|---|---|---|---|---|---|
| Baseline | ✗ | ✗ | ✗ | 55.7 | 21.1 | 37.0 | 45.0 | 76.7 | 8.7 |
| Uni-Walker w/o KIS | ✗ | ✓ | ✓ | 60.3 | 14.2 | 50.2 | 23.9 | 77.6 | 7.7 |
| Uni-Walker w/o SSC | ✓ | ✓ | ✗ | 59.7 | 15.1 | 44.7 | 30.6 | 77.9 | 7.3 |
| Uni-Walker w/o ECAS | ✓ | ✗ | ✓ | 58.1 | 17.4 | 44.7 | 32.3 | 78.3 | 6.9 |
| **Our Uni-Walker** | ✓ | ✓ | ✓ | **67.3** | **4.3** | **62.3** | **5.7** | **81.3** | **3.5** |

Table 5: Ablation study results (%) for the methods with **Task Specific Knowledge Exploration**.

| Methods | ESOC | NSCoT | SR ↑ | SR-F ↓ | SPL ↑ | SPL-F ↓ | OSR ↑ | OSR-F ↓ |
|---|---|---|---|---|---|---|---|---|
| Baseline | ✗ | ✗ | 49.0 | 29.2 | 33.9 | 45.0 | 72.3 | 14.0 |
| Uni-Walker w/o ESOC | ✗ | ✓ | 63.5 | 9.8 | 60.6 | 8.2 | 79.7 | 5.3 |
| Uni-Walker w/o NSCoT | ✓ | ✗ | 51.1 | 27.3 | 35.5 | 46.3 | 75.3 | 10.5 |
| **Our Uni-Walker** | ✓ | ✓ | **67.3** | **4.3** | **62.3** | **5.7** | **81.3** | **3.5** |

Table 6: Ablation study results (%) for the **Task-Aware Knowledge Aggregation** methods.

| Methods | IM | OM | MM | SR ↑ | SR-F ↓ | SPL ↑ | SPL-F ↓ | OSR ↑ | OSR-F ↓ |
|---|---|---|---|---|---|---|---|---|---|
| Uni-Walker w IM | ✓ | ✗ | ✗ | 35.0 | 50.1 | 23.2 | 65.0 | 46.6 | 49.5 |
| Uni-Walker w OM | ✗ | ✓ | ✗ | 65.1 | 9.6 | **62.7** | 7.5 | 80.1 | 5.5 |
| **Our Uni-Walker** | ✗ | ✗ | ✓ | **67.3** | **4.3** | 62.3 | **5.7** | **81.3** | **3.5** |

navigation knowledge is preserved while task-specific expertise remains disentangled rather than entangled or overwritten; Dynamic knowledge aggregation through TAKA, which activates the most relevant experts for unseen scenes, to reuse transferable skill knowledge when encountering unseen scenes. Together, these mechanisms allow Uni-Walker to adapt to new tasks while retaining previously learned knowledge, leading to significantly better performance in unseen generalization.

## 4.3 ABLATION STUDIES

**How Does Navigation Tasks Shared Knowledge Perform?** The ablation studies results of the exploration for navigation tasks shared knowledge are summarized in Table 4. Specifically, the "KIS" denotes the proposed Knowledge Inheritance Strategy, which summarizes previously learned knowledge and utilizes it for learning new tasks. Removing the inheritance leads to slower convergence and weaker transfer to new tasks. The "ECAS" denotes the proposed Experts Co-Activation Strategy, which directly utilizes previously acquired knowledge to accomplish the current task. Removing the co-activation leads to a reduction in the exploitation of previous knowledge. The "SSC" denotes the proposed shared smoothing consolidation loss for shared subspace $\mathbf{A}'$, which facilitates the smooth updating of shared subspaces during the continual learning process. Removing the loss leads to overfitting new knowledge and the catastrophic forgetting of old knowledge. Based on the ablation results, the proposed methods enable Uni-Walker to explore and exploit task-shared knowledge between diverse tasks, effectively improving new task learning and reducing catastrophic forgetting.

**How Does Navigation Tasks Specific Knowledge Perform?** These ablation studies' results about the exploration for navigation tasks shared knowledge are summarized in the Table 5. Specifically, the "ESOC" denotes the proposed Expert Subspace Orthogonality Constraint on $\mathbf{B}$, which facilitates the independence of each expert subspace to learn task-specific knowledge. Removing the orthogonality constraint may lead to overlapping of the expert subspaces, resulting in the failure of knowledge decoupling. The "NSCoT" denotes the proposed Navigation Specific Chain of Thought, which provides specific reasoning processes for each specific navigation task. Removing NSCoT and applying a fixed reasoning template (prompting only the reasoning user instructions to complete navigation tasks) leads to a decline in complex task performance. Based on the ablation results, the proposed methods enable Uni-Walker to explore and exploit task-specific knowledge between diverse navigation tasks, effectively improving the new specific navigation task learning efficiency.

**How Does Task-Aware Knowledge Aggregation Perform?** These ablation studies' results about the Task-Aware Knowledge Aggregation are summarized in the Table 6. Specifically, the "IM" denotes matching with user instruction; the "OM" denotes matching with agent observation; the "MM"

denotes our proposed mixed matching method. Based on the ablation results, the proposed aggregation method comprehensively considers both matching modes and achieves the best performance.

**How does computational complexity scale as the number of tasks increases?** We analyze the resource cost of LoRA and the Fisher Information Matrix in our DE-LoRA architecture. For each additional task, Uni-Walker adds: $\approx 2.1$ MB for the LoRA expert subspace and maintains a Fisher Matrix of the same size. Thus, even when the number of tasks grows beyond 100, the total storage remains modest: $100 \times 4.2$ MB $\approx 0.4$ GB. Such overhead is negligible for modern LLM-based navigation systems (7B, 13B, or larger), demonstrating the practical scalability of the architecture.

## 5 RELATED WORKS

**Embodied Navigation.** Embodied navigation requires an agent to follow user language instructions and reach a goal in a visual scene Liu et al. (2025); Han et al. (2025); Bar et al. (2025); Nie et al. (2025). Recent research has established three representative tasks. Vision-and-Language Navigation (VLN) Anderson et al. (2018a) requires user step-by-step grounding of detailed instructions, and has been widely studied in both discrete Anderson et al. (2018b); Ku et al. (2020); Qi et al. (2020b); Fu et al. (2020); Hong et al. (2020) and continuous environments Krantz et al. (2020); Raychaudhuri et al. (2021); Chen et al. (2020; 2022). Object Localization Navigation (OLN) Qi et al. (2020a) requires the agent to localize a distant object given concise referring expressions, emphasizing high-level semantic understanding Arnaud et al. (2025); Yu & Saniie (2025); Lei et al. (2025). Dialog-based Understanding Navigation (DUN) Thomason et al. (2020) extends this paradigm to interactive settings, where agents need to interpret multi-turn dialogues to reason user intent Majumdar et al. (2025); Qiao et al. (2025). Building upon these advances, recent efforts aim to construct universal navigation agents by jointly training across multiple tasks, leveraging large pretrained LLMs Zheng et al. (2024); Zhou et al. (2024); Qiao et al. (2025); Gao et al. (2025a); Zhang et al. (2025b). However, these methods typically assume fixed task distributions and are prone to catastrophic forgetting when adapting to new scenes or instruction styles. This motivates the study of lifelong navigation learning, where agents continually integrate knowledge from sequential tasks without forgetting.

**Continual Learning.** Continual learning seeks to enable models to acquire new skills over time without erasing previously learned ones Li et al. (2025); Yang et al. (2025); Wang et al. (2024; 2026a). Existing approaches can be broadly grouped into three categories. Parameter regularization methods constrain weight updates to preserve important parameters for past tasks Rebuffi et al. (2017); Li & Hoiem (2017); Derakhshani et al. (2021); Douillard et al. (2020). Architecture-based methods dedicate specific network components to different sequential tasks Jung et al. (2020); Wu et al. (2021); Wang et al. (2022); Toldo & Ozay (2022). Replay-based methods store or generate data from past tasks to rehearse old skills Bang et al. (2021); Rebuffi et al. (2017); Sun et al. (2022); Wan et al. (2024); Li et al. (2022); Xiang et al. (2019), though such approaches can be memory-intensive and raise privacy concerns. In robotics, continual learning has been explored to enable agents to expand their skill repertoire through lifelong interaction Meng et al. (2025); Yao et al. (2025); Zhu et al. (2025); Ayub et al. (2025b;a). Despite progress, these existing techniques rarely address the unique challenges of embodied navigation, where both scene diversity and instruction variability necessitate balancing shared knowledge transfer and task-specific reasoning.

## 6 CONCLUSION

In this work, we introduce Lifelong Embodied Navigation Learning (LENL), a novel task that enables a navigation agent to continually learn new multiple navigation tasks, including new scenes and with new user instruction styles. We also propose Uni-Walker, a lifelong embodied navigation framework that decouples navigation knowledge into task-shared and task-specific components with Decoder Extension LoRA (DE-LoRA). For the shared knowledge, we design a Knowledge Inheritance Strategy and Experts Co-Activation Strategy to facilitate shared knowledge transfer and refinement across multiple navigation tasks. For the specific knowledge, we propose an Expert Subspace Orthogonality Constraint together and a Navigation-Specific Chain-of-Thought reasoning mechanism to capture task-specific knowledge and enhance specific instruction styles understanding. Extensive experiments are performed to demonstrate the effectiveness and superiority of proposed Uni-Walker.

## 7 ACKNOWLEDGMENTS

This work was supported by the National Natural Science Foundation of China under Grant T2596040, T2596045 and U23A20343, CAS Project for Young Scientists in Basic Research, Grant YSBR-041, Liaoning Provincial "Selecting the Best Candidates by Opening Competition Mechanism" Science and Technology Program under Grant 2023JH1/10400045, Joint Innovation Fund of DICP & SIA under Grant UN202401, Fundamental Research Project of SIA under Grant 2024JC3K01, Natural Science Foundation of Liaoning Province under Grant 2025-BS-0193.

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

# APPENDIX

# A    EXPERIMENTS IMPLEMENTATION

Below we provide a detailed experiment implementation.

**Benchmark and task setup**. We evaluate on the Lifelong Embodied Navigation (LENL) benchmark built on the Matterport3D simulator Anderson et al. (2018a). The benchmark contains 18 distinct navigation scenes and three instruction styles: (1) VLN (step-by-step route instructions), (2) OLN (object-localization style instructions), and (3) DUN (dialogue-style instructions). Tasks are presented as an ordered sequence of 18 tasks: the first 15 tasks are used for continual (lifelong) learning while the final 3 tasks are reserved for zero-shot generalization tests in unseen environments. In the LENL protocol the task identifier $t$ is available to the training procedure (to allow task-specific adapter/expert creation) but is *not* provided at test time (evaluation is task-agnostic). The scene splits and examples are illustrated in Fig. 5 of the main paper. The specific statistical information for the 18 tasks is shown in Table 7.

Table 7: Statistics of the 18 sequential tasks of LENL benchmark. Each task corresponds to a unique scene under one instruction style.

| Task ID | Type | Scene ID | INST Style | Train INST Number | Test INST Number |
|---------|------|----------|------------|-------------------|------------------|
| S1 | Continual Learning | 1LXtFkjw3qL | VLN | 801 | 99 |
| S2 | Continual Learning | 1pXnuDYAj8r | VLN | 813 | 90 |
| S3 | Continual Learning | pRbA3pwrgk9 | OLN | 336 | 195 |
| S4 | Continual Learning | 5q7pvUzZiYa | VLN | 738 | 63 |
| S5 | Continual Learning | 759xd9YjKW5 | OLN | 693 | 147 |
| S6 | Continual Learning | VVfe2KiqLaN | DUN | 255 | 24 |
| S7 | Continual Learning | 82sE5b5pLXE | VLN | 837 | 63 |
| S8 | Continual Learning | p5wJjkQkbXX | OLN | 1671 | 444 |
| S9 | Continual Learning | B6ByNegPMKs | VLN | 801 | 72 |
| S10 | Continual Learning | 7y3sRwLe3Va | OLN | 291 | 153 |
| S11 | Continual Learning | D7N2EKCX4Sj | VLN | 837 | 63 |
| S12 | Continual Learning | mJXqzFtmKg4 | DUN | 228 | 36 |
| S13 | Continual Learning | ULsKaCPVFJR | OLN | 1848 | 75 |
| S14 | Continual Learning | cV4RVeZvu5T | VLN | 837 | 63 |
| S15 | Continual Learning | rPc6DW4iMge | OLN | 654 | 141 |
| S16 | Unseen Generalization | EDJbREhghzL | VLN | – | 72 |
| S17 | Unseen Generalization | V2XKFyX4ASd | OLN | – | 69 |
| S18 | Unseen Generalization | VzqfbhrpDEA | DUN | – | 27 |

**Comparisons.** All methods (our method and comparisons) use the same NavLLM backbone to ensure a fair comparison. Following NavLLM (Zheng et al. (2024)), visual encodings are produced by a CLIP visual encoder (Radford et al. (2021)) and fused into the language model i.e., Vicuna (Chiang et al. (2023)), via the same multi-modal interface used across experiments. Implementation details of the multimodal interface follow the main paper (§2). For a fair and informative comparison we re-implemented or configured a collection of state-of-the-art LoRA-based continual learning and Mixture-of-Experts (MoE) methods on the same NavLLM backbone, input preprocessing, training schedule and evaluation pipeline. All baselines train only adapter/LoRA-style parameters while keeping the backbone model weights frozen unless otherwise stated. Below we summarize each baseline, the key implementation choices used in our experiments, and their inference behavior.

- **Seq-FT.** Sequential fine-tuning of the base model using a *vanilla* LoRA adapter per task. For each task we train a standalone LoRA module (rank $r = 64$). No explicit mechanism is used to avoid forgetting; after finishing the whole curriculum the stored LoRA module corresponding to a requested task is used for inference. Seq-FT serves as a lower-bound reference for catastrophic forgetting.

- **LwF-LoRA** Li & Hoiem (2017). Extension of the Learning-without-Forgetting paradigm to LoRA: during training on a new task we apply knowledge-distillation losses to a frozen copy of the model (logits and selected hidden states) so as to encourage the newly-trained LoRA to preserve behavior on previously seen tasks. We follow the distillation configura-

tion and hyper-parameterization recommended in Li & Hoiem (2017). At inference time the stored LoRA corresponding to the target task is loaded (rank $r = 64$).

- **EWC-LoRA** Xiang et al. (2023). Elastic Weight Consolidation adapted to LoRA parameters: after each finished task we estimate parameter importance (Fisher information) for the trained LoRA parameters and penalize future updates to parameters with high importance. We apply the EWC penalty only to LoRA adapter weights and follow the hyperparameter schedule described in Xiang et al. (2023). The stored per-task LoRA adapter is used for evaluation after training (rank $r = 64$).

- **Dense MoLE** Chen et al. (2024). A dense Mixture-of-LoRA-Experts design in which all experts are activated for every forward pass (i.e., dense combination of expert outputs). This design captures cross-task interactions more richly but increases per-step computation and memory usage compared to sparse routing. Expert outputs are aggregated with learnable gating coefficients; the full MOE-LoRA set of adapters is retained for inference (rank $r = 16$, $K = 8$).

- **Sparse MoLE** Dou et al. (2024). Sparse routing MoE applied to LoRA adapters: an input is routed to only a subset of experts to reduce computation and encourage specialization. We use a top-$k$ routing policy with $k = 2$ experts per instance, implement load-balancing auxiliary losses where applicable, and keep gating parameters trainable. After training the full sparse MOE-LoRA checkpoint (all expert adapters and the gating network) is used for inference (rank $r = 16$, $K = 8$).

- **MoLA** Gao et al. (2024). Builds on Sparse MoLE by introducing hierarchical/deeper expert layers for richer adaptation. Concretely, we instantiate a two-level expert hierarchy with 4 shallow experts and 16 deep experts (shallow experts feed into deeper expert layers). Routing is performed at both levels; during inference the trained MoLA adapter collection and gating modules are used (rank $r = 16$.

- **O-LoRA** Wang et al. (2023a). Per-task vanilla LoRA storage combined with an orthogonality-promoting regularizer applied during training to encourage disentangled task representations. Practically this requires storing one LoRA module per task. For inference we adopt the proposed TAKA selection method. (rank $r = 24$.

- **HydraLoRA** Tian et al. (2024). A hybrid design that shares a global module $\mathbf{A}$ across all tasks to capture common knowledge while using multiple task-specific $\mathbf{B}$ modules for specialization. During each task's training only the corresponding $\mathbf{B}$ (and optionally $\mathbf{A}$) are updated according to the method specification. At inference the composed HydraLoRA adapter ($\mathbf{A}$ combined with selected $\mathbf{B}$) is loaded (rank $r = 16$, $K = 8$).

- **BranchLoRA** Zhang et al. (2025a). Extends sparse routing with an explicit branching mechanism that allows the model to select different $\mathbf{B}$ branches for different input modes or instruction styles. Branch selection is learned, and branch-specific LoRA adapters are stored per branch. For evaluation, BranchLoRA uses branch selection to pick the appropriate $\mathbf{B}$ at test time (rank $r = 16$, $K = 8$).

- **SD-LoRA** Wu et al. (2025). Stores $T$ task-specific LoRA modules and dynamically composes them during inference to synthesize behavior for the current input. We follow the composition rules and hyper-parameter choices recommended in Wu et al. (2025). For inference we adopt the proposed TAKA selection method. (rank $r = 24$.

**Evaluation metrics.** Let an agent's executed trajectory within an episode be denoted by $\mathcal{T} = (\mathbf{p}_1, \mathbf{p}_2, \ldots, \mathbf{p}_n)$, where $\mathbf{p}_i$ is the agent's position at timestep $i$. Let $\mathbf{g}$ denote the goal position and $d(\cdot, \cdot)$ be the distance function in the environment (e.g., geodesic or Euclidean distance depending on the simulator). We use three standard navigation metrics described below. In all experiments we adopt a success threshold $\epsilon$ (we set $\epsilon = 1.0$ m unless stated otherwise).

**Success rate (SR).** An episode is counted as successful if the final agent position is within $\epsilon$ of the goal. Formally, the per-episode success indicator is

$$\mathrm{SR}_{\mathrm{ep}} = \begin{cases} 1, & \text{if } d(\mathbf{p}_n, \mathbf{g}) \leq \epsilon, \\ 0, & \text{otherwise.} \end{cases} \tag{12}$$

The reported SR is the mean of $\text{SR}_{\text{ep}}$ over all test episodes. **Oracle success rate (OSR).** Oracle success measures whether the agent visited a position within $\epsilon$ of the goal at any time during the episode. It is defined per episode as

$$\text{OSR}_{\text{ep}} = \begin{cases} 1, & \text{if } \min_{1 \leq i \leq n} d(\mathbf{p}_i, \mathbf{g}) \leq \epsilon, \\ 0, & \text{otherwise,} \end{cases} \tag{13}$$

and the dataset-level OSR is the average of $\text{OSR}_{\text{ep}}$ across episodes.

**Success weighted by Path Length (SPL).** SPL jointly measures success and path efficiency. For episode $j$, let $S_j \in \{0, 1\}$ be the success indicator (as above), let $L_j$ denote the length of the agent's executed path, and let $L_j^*$ denote the shortest-path (optimal) distance from the start to the goal. The standard per-episode contribution for SPL is

$$\text{SPL}_{\text{ep}} = S_j \cdot \frac{L_j^*}{\max(L_j, L_j^*)}, \tag{14}$$

and the reported SPL is the mean of $\text{SPL}_{\text{ep}}$ over all episodes. SPL penalizes long or inefficient trajectories while counting only successful attempts.

We use eight NVIDIA RTX 6000 Ada Generation GPUs with PyTorch 2.1.2 (cu121) for training and testing. And the Adam optimizer with an initial learning rate of $3.0 \times 10^{-5}$ is used for training. The low-rank $r = 16$ of LORA, the number of experts is $k = 2$, $\mu = 0.5$, and all the other hyper-parameter are consistent with the NavLLM (Zheng et al. (2024)). We summarize some important parameter settings as shown in Table 8.

Table 8: Summary of primary hyper-parameters used in experiments.

| Hyper-parameter | Value |
| --- | --- |
| TOP-K activated expert | $K = 2$ |
| LoRA rank | $r = 16$ |
| balance hyperparameter of $\mathcal{L}_{ssc,t}$ | $\lambda_{ssc} = 0.1$ |
| balance hyperparameter of $\mathcal{L}_{esoc,t}$ | $\lambda_{esoc} = 0.1$ |
| perturbation for avoiding the computational avalanches | $\epsilon = 0.01$ |
| Fisher smoothing | $\omega = 0.9$ |
| elements threshold | $\mu = 0.5$ |
| LLM | Vicuna-7B-v0 |
| ViT | EVA-CLIP-02-Large (428M) |
| training steps | 2000 |
| batch size | 64 |
| sampling strategy with a temperature | 0.01 |

## B    ALGORITHM SUMMARY

For ease of readers understanding, a summary of the proposed Uni-Walker training algorithm is provided in the Algorithm 1, and a summary of inference algorithm is provided in the Algorithm 2.

## C    DETAILED COMPARISON RESULTS

Fig. 6 presents the average results for SPL, SPL-F, OSR, and OSR-F. In this section, we provide the task-wise comparison results. The original comparison results (SPL, F-SPL, OSR, F-OSR) for Fig. 6 are summarized in Table 9, Table 10, Table 11, and Table 12. Uni-Walker achieves a higher average success rate of 66%, surpassing the previous best (59%) by 7%, and a lower forgetting rate of 5%, improving the prior best (16%) by 11%. Uni-Walker achieves a higher SPL of 61%, surpassing the previous best (38%) by 23%, and a lower forgetting rate of 7%, improving the prior best (42%) by 35%. It achieves a higher OSR of 81%, surpassing the previous best (79%) by 2%, and a lower forgetting rate of 5%, improving the prior best (7%) by 2%. Based on the results, our Uni-Walker achieves consistent superiority across various metrics. The visualization examples of VLN, OLN, DUN are visualized in Fig. 7.

---

**Algorithm 1** Uni-Walker: Lifelong Training

---

**Require:** Frozen backbone $F_{\theta_0}$, initialize shared $\mathbf{A}$, initial expert set $\{\mathbf{B}_1\}$, task sequence $\{\mathcal{T}_1, \ldots, \mathcal{T}_T\}$, TAKA index $\mathcal{R}_e = \emptyset$
1: **for** task $t = 1$ **to** $T$ **do**
2:    **if** $t > 1$ **then**
3:       Create new expert $\mathbf{B}_t$
4:       Initialize $\mathbf{B}_t$ via Knowledge Inheritance Strategy (KIS) with E.q2
5:    **end if**
6:    Add $(E_{S,t}, E_{I,t})$ placeholder to retrieval index $\mathcal{R}_e$
7:    Compute $t$-th Fisher Matrix $F_t$ with Eq.5.
8:    Update and save the $t$-th Fisher Matrix $F_t$ with Eq.6.
9:    **for** epoch = 1 **to** max_epochs$_t$ **do**
10:      **for** each minibatch $(\mathcal{O}, \mathcal{I}, \mathcal{A})$ from task $T_t$ **do**
11:         $E_o \leftarrow \text{CLIP\_V}(\mathcal{O})$,    $E_i \leftarrow \text{CLIP\_T}(\mathcal{I})$
12:         Activating TOP-K expert subspace $\mathcal{K} \leftarrow \text{TAKA}(E_o, E_i, \mathcal{R}_e, \mu)$ with E.q10
13:         Compute $\mathcal{L}_{ssc,t}$ (shared smoothing consolidation loss) using Fisher matrices with Eq.4
14:         Compute $\mathcal{L}_{esoc,t}$ (expert subspace orthogonality constrain loss) with Eq.7
15:         Compute $\mathcal{L}_t$ (auto-regressive action generation loss) with Eq.8
16:         Back-propagate and update only $\mathbf{A}$ and $\mathbf{B}_t$ parameters (freeze other $\mathbf{B}_n$)
17:      **end for**
18:    **end for**
19:    Save expert checkpoint $\mathbf{B}_t$ and shared $\mathbf{A}$ checkpoint
20: **end for**
21: **return** saved $\{\mathbf{B}_1, \ldots, \mathbf{B}_T\}$, final $\mathbf{A}$, retrieval index $\mathcal{R}_e$

---

**Algorithm 2** Uni-Walker: Inference with TAKA and Expert Co-activation

---

**Require:** Frozen backbone $F_{\theta_0}$, shared $\mathbf{A}$, saved experts $\{\mathbf{B}_1, \ldots, \mathbf{B}_T\}$, retrieval index $\mathcal{R}_e$, input $(\mathcal{O}_q, \mathcal{I}_q)$, threshold $\mu$, top-K
**Ensure:** Predicted action token sequence
1: $E_o \leftarrow \text{CLIP\_V}(\mathcal{O}_q)$,    $E_i \leftarrow \text{CLIP\_T}(\mathcal{I}_q)$
2: Compute instruction similarities $s_{i,t} = \text{cosine}(E_i, E_{I,t})$ for all $(\cdot, E_{I,t}) \in \mathcal{R}_e$ with Eq.9
3: Build instruction mask $M_t = \mathbf{1}\{s_{i,t} \geq \mu\}$
4: Compute observation similarities $s_{o,t} = \text{cosine}(E_o, E_{S,t})$, then $\tilde{s}_t = M_t \cdot s_{o,t}$ with Eq.9
5: Select top-K experts $\mathcal{K} = \text{TopK}(\tilde{s}_t, K)$
6: Forward using $\mathcal{F}_{\theta_0}$ with adapters combined over $\mathcal{K}$ with Eq.3
7: Generate tokens to produce next actions using the model's policy head
8: **return** action token sequence

---

Table 9: Test results (task-wise **SPL** ↑, %) of comparison experiment with our LENL settings.

| Comparison Methods | S1 | S2 | S3 | S4 | S5 | S6 | S7 | S8 | S9 | S10 | S11 | S12 | S13 | S14 | S15 | S16 | S17 | S18 | Avg. |
|---|---|---|---|---|---|---|---|---|---|---|---|---|---|---|---|---|---|---|---|
| Seq-FT (sequential fine-tuning) | 0 | 0 | 0 | 2 | 2 | 0 | 2 | 10 | 6 | 6 | 8 | 15 | 10 | 12 | 79 | 0 | 0 | 0 | 8 |
| LwF-LoRA (Li & Hoiem (2017)) | 0 | 0 | 0 | 0 | 2 | 0 | 2 | 11 | 5 | 4 | 6 | 21 | 36 | 24 | 80 | 0 | 0 | 0 | 11 |
| EWC-LoRA (Xiang et al. (2023)) | 0 | 2 | 0 | 4 | 0 | 0 | 0 | 10 | 6 | 2 | 4 | 25 | 32 | 30 | 80 | 0 | 0 | 0 | 11 |
| Dense MoLE (Chen et al. (2024)) | 2 | 2 | 2 | 4 | 4 | 0 | 2 | 10 | 8 | 12 | 10 | 20 | 36 | 40 | 78 | 2 | 0 | 2 | 13 |
| Sparse MoLE (Dou et al. (2024)) | 2 | 2 | 0 | 10 | 2 | 0 | 10 | 13 | 12 | 15 | 10 | 24 | 34 | 42 | 78 | 0 | 2 | 0 | 14 |
| MoLA (Gao et al. (2024)) | 4 | 6 | 0 | 6 | 2 | 5 | 12 | 12 | 13 | 11 | 15 | 23 | 36 | 44 | 80 | 6 | 6 | 0 | 16 |
| HydraLoRA (Tian et al. (2024)) | 8 | 10 | 12 | 13 | 2 | 10 | 12 | 16 | 8 | 12 | 16 | 22 | 37 | 45 | 80 | 14 | 11 | 10 | 19 |
| BranchLoRA (Zhang et al. (2025a)) | 8 | 10 | 13 | 15 | 4 | 12 | 15 | 19 | 15 | 16 | 18 | 27 | 41 | 36 | 78 | 15 | 10 | 11 | 20 |
| O-LoRA (Wang et al. (2023a))+TAKA | 30 | 31 | 32 | 45 | 26 | 24 | 34 | 36 | 25 | 34 | 35 | 26 | 65 | 48 | 80 | 32 | 32 | 31 | 37 |
| SD-LoRA (Wu et al. (2025))+TAKA | 32 | 33 | 34 | 35 | 26 | 26 | 36 | 38 | 26 | 36 | 37 | 26 | 66 | 49 | 80 | 38 | 30 | 42 | 38 |
| **Uni-Walker (ours)** | **66** | **63** | **59** | **61** | **49** | **50** | **65** | **74** | **50** | **72** | **56** | **51** | **82** | **56** | **80** | **71** | **55** | **45** | **61** |

## D   DETAILED ABLATION RESULTS

We perform three ablation about our Uni-Walker in the main paper. In this section, we provide the task-wise ablation results. The ablation studies results of the exploration for navigation tasks shared knowledge are summarized in Table 4, and the task-wise ablation results (SR) are summarized in

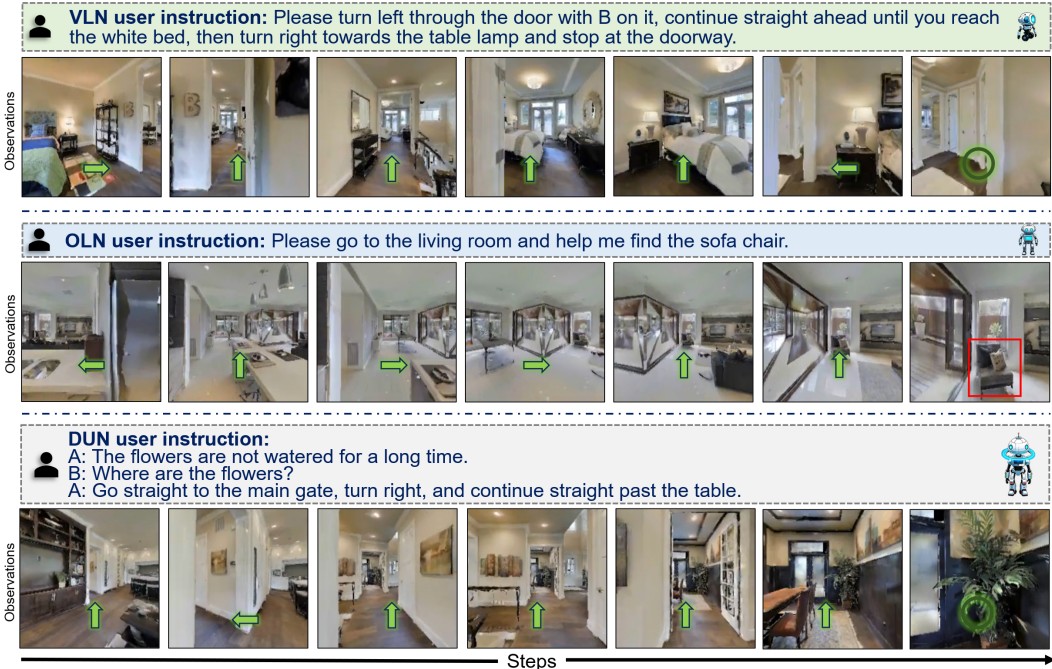

Figure 7: Illustration of three visualization examples of VLN, OLN, DUN. We visualize each step of the Uni-Walker's selection process and annotate it with arrows to clearly illustrate the navigation sequence.

Table 10: Test results (task-wise **SPL-F** ↓, %) of comparison experiment with our LENL settings.

| Comparison Methods | S1 | S2 | S3 | S4 | S5 | S6 | S7 | S8 | S9 | S10 | S11 | S12 | S13 | S14 | S15 | S16 | S17 | S18 | Avg. |
|---|---|---|---|---|---|---|---|---|---|---|---|---|---|---|---|---|---|---|---|
| Seq-FT (sequential fine-tuning) | 100 | 100 | 100 | 97 | 96 | 100 | 97 | 88 | 88 | 92 | 86 | 72 | 88 | 79 | 0 | 100 | 100 | 100 | 88 |
| LwF-LoRA (Li & Hoiem (2017)) | 100 | 100 | 100 | 100 | 96 | 100 | 97 | 87 | 90 | 95 | 90 | 61 | 57 | 58 | 0 | 100 | 100 | 100 | 85 |
| EWC-LoRA (Xiang et al. (2023)) | 100 | 97 | 100 | 94 | 100 | 100 | 100 | 88 | 88 | 97 | 93 | 54 | 61 | 47 | 0 | 100 | 100 | 100 | 84 |
| Dense MoLE (Chen et al. (2024)) | 97 | 97 | 97 | 94 | 92 | 100 | 97 | 88 | 85 | 84 | 83 | 63 | 57 | 30 | 0 | 98 | 100 | 96 | 81 |
| Sparse MoLE (Dou et al. (2024)) | 97 | 97 | 100 | 85 | 96 | 100 | 85 | 85 | 77 | 79 | 83 | 56 | 60 | 26 | 0 | 100 | 97 | 100 | 79 |
| MoLA (Gao et al. (2024)) | 94 | 91 | 100 | 91 | 96 | 90 | 82 | 86 | 75 | 85 | 75 | 57 | 57 | 24 | 0 | 93 | 91 | 100 | 77 |
| HydraLoRA (Tian et al. (2024)) | 88 | 85 | 81 | 80 | 96 | 81 | 82 | 81 | 85 | 84 | 73 | 61 | 56 | 22 | 0 | 85 | 84 | 79 | 72 |
| BranchLoRA (Zhang et al. (2025a)) | 88 | 85 | 79 | 77 | 92 | 77 | 78 | 78 | 72 | 79 | 69 | 53 | 51 | 38 | 0 | 84 | 85 | 77 | 70 |
| O-LoRA (Wang et al. (2023a))+TAKA | 56 | 52 | 48 | 31 | 50 | 54 | 51 | 58 | 53 | 55 | 41 | 54 | 23 | 17 | 0 | 65 | 53 | 35 | 44 |
| SD-LoRA (Wu et al. (2025))+TAKA | 53 | 49 | 45 | 46 | 50 | 50 | 47 | 55 | 51 | 52 | 37 | 54 | 21 | 16 | 0 | 58 | 56 | 13 | 42 |
| **Uni-Walker (ours)** | **3** | **3** | **5** | **6** | **6** | **4** | **6** | **13** | **6** | **4** | **5** | **11** | **5** | **3** | **0** | **22** | **19** | **8** | **7** |

Table 11: Test results (task-wise **OSR** ↑, %) of comparison experiment with our LENL settings.

| Comparison Methods | S1 | S2 | S3 | S4 | S5 | S6 | S7 | S8 | S9 | S10 | S11 | S12 | S13 | S14 | S15 | S16 | S17 | S18 | Avg. |
|---|---|---|---|---|---|---|---|---|---|---|---|---|---|---|---|---|---|---|---|
| Seq-FT (sequential fine-tuning) | 12 | 12 | 14 | 16 | 18 | 16 | 17 | 19 | 16 | 18 | 15 | 35 | 36 | 36 | 100 | 16 | 15 | 15 | 24 |
| LwF-LoRA (Li & Hoiem (2017)) | 12 | 15 | 16 | 15 | 13 | 15 | 16 | 29 | 16 | 32 | 32 | 45 | 55 | 51 | 100 | 13 | 15 | 16 | 28 |
| EWC-LoRA (Xiang et al. (2023)) | 13 | 15 | 16 | 15 | 15 | 16 | 15 | 29 | 15 | 34 | 36 | 46 | 56 | 52 | 100 | 16 | 18 | 17 | 29 |
| Dense MoLE (Chen et al. (2024)) | 21 | 22 | 18 | 23 | 16 | 18 | 19 | 31 | 16 | 33 | 41 | 48 | 69 | 68 | 100 | 22 | 20 | 17 | 33 |
| Sparse MoLE (Dou et al. (2024)) | 23 | 23 | 19 | 25 | 19 | 21 | 23 | 36 | 19 | 26 | 40 | 47 | 69 | 72 | 100 | 25 | 26 | 18 | 35 |
| MoLA (Gao et al. (2024)) | 26 | 24 | 16 | 24 | 19 | 23 | 25 | 37 | 21 | 29 | 45 | 48 | 69 | 73 | 100 | 28 | 22 | 26 | 36 |
| HydraLoRA (Tian et al. (2024)) | 27 | 25 | 21 | 25 | 20 | 25 | 24 | 36 | 22 | 27 | 46 | 48 | 72 | 68 | 100 | 29 | 26 | 22 | 37 |
| BranchLoRA (Zhang et al. (2025a)) | 32 | 29 | 23 | 26 | 24 | 26 | 28 | 36 | 26 | 31 | 46 | 51 | 88 | 73 | 100 | 33 | 30 | 31 | 41 |
| O-LoRA (Wang et al. (2023a))+TAKA | 76 | 81 | 80 | 64 | 82 | 82 | 71 | 82 | 67 | 70 | 62 | 86 | 98 | 80 | 100 | 77 | 68 | 69 | 77 |
| SD-LoRA (Wu et al. (2025))+TAKA | 76 | 82 | 85 | 66 | 88 | 84 | 76 | 85 | 67 | 72 | 66 | **85** | 98 | **81** | 100 | 79 | 68 | 71 | 79 |
| **Uni-Walker (ours)** | **82** | **87** | **88** | **67** | **90** | **90** | **72** | **83** | **63** | **73** | **62** | 84 | **100** | 78 | **100** | **84** | **76** | **79** | **81** |

Table 13, and the task-wise ablation results (F-SR) are summarized in Table 14, and the task-wise ablation results (SPL) are summarized in Table 15, and the task-wise ablation results (F-SPL) are summarized in Table 16, and the task-wise ablation results (OSR) are summarized in Table 17, and the task-wise ablation results (F-OSR) are summarized in Table 18. The ablation studies' results about the exploration for navigation tasks shared knowledge are summarized in the Table 5, and

Table 12: Test results (task-wise **OSR-F** ↓, %) of comparison experiment with our LENL settings.

| Comparison Methods | S1 | S2 | S3 | S4 | S5 | S6 | S7 | S8 | S9 | S10 | S11 | S12 | S13 | S14 | S15 | S16 | S17 | S18 | Avg. |
|---|---|---|---|---|---|---|---|---|---|---|---|---|---|---|---|---|---|---|---|
| Seq-FT (sequential fine-tuning) | 86 | 87 | 84 | 77 | 80 | 83 | 77 | 78 | 75 | 76 | 77 | 59 | 64 | 58 | 0 | 83 | 83 | 83 | 73 |
| LwF-LoRA (Li & Hoiem (2017)) | 86 | 83 | 82 | 79 | 86 | 84 | 80 | 66 | 75 | 57 | 54 | 48 | 45 | 40 | 0 | 86 | 83 | 81 | 67 |
| EWC-LoRA (Xiang et al. (2023)) | 85 | 83 | 82 | 79 | 84 | 83 | 81 | 66 | 77 | 55 | 48 | 47 | 44 | 39 | 0 | 83 | 79 | 80 | 66 |
| Dense MoLE (Chen et al. (2024)) | 75 | 75 | 80 | 67 | 83 | 80 | 76 | 64 | 75 | 56 | 41 | 44 | 31 | 20 | 0 | 77 | 77 | 80 | 61 |
| Sparse MoLE (Dou et al. (2024)) | 73 | 74 | 79 | 64 | 79 | 77 | 71 | 65 | 42 | 45 | 31 | 15 | 0 | 74 | 79 | 79 | 59 |
| MoLA (Gao et al. (2024)) | 69 | 73 | 82 | 66 | 79 | 75 | 68 | 56 | 68 | 61 | 35 | 44 | 31 | 14 | 0 | 71 | 74 | 70 | 58 |
| HydraLoRA (Tian et al. (2024)) | 68 | 72 | 77 | 64 | 78 | 73 | 70 | 58 | 68 | 64 | 33 | 44 | 28 | 20 | 0 | 69 | 70 | 74 | 57 |
| BranchLoRA (Zhang et al. (2025a)) | 62 | 67 | 74 | 63 | 74 | 72 | 65 | 58 | 62 | 59 | 33 | 41 | 12 | 14 | 0 | 65 | 65 | 64 | 53 |
| O-LoRA (Wang et al. (2023a))+TAKA | 11 | 9 | 11 | 9 | 11 | 11 | 10 | 4 | 3 | 7 | 10 | 0 | 2 | 6 | 0 | 19 | 21 | 20 | 9 |
| SD-LoRA (Wu et al. (2025))+TAKA | 11 | 8 | 6 | 6 | 4 | 9 | 4 | 0 | 3 | 4 | 4 | 1 | 2 | 5 | 0 | 17 | 21 | 17 | 7 |
| **Uni-Walker (ours)** | **4** | **2** | **2** | **4** | **2** | **2** | **8** | **2** | **7** | **3** | **5** | **2** | **0** | **8** | **0** | **12** | **12** | **8** | **5** |

Table 13: Ablation study results (**SR** ↑, %) for the methods with task-shared navigation knowledge exploration.

| Methods | S1 | S2 | S3 | S4 | S5 | S6 | S7 | S8 | S9 | S10 | S11 | S12 | S13 | S14 | S15 | Avg. |
|---|---|---|---|---|---|---|---|---|---|---|---|---|---|---|---|---|
| Baseline | 50 | 56 | 46 | 51 | 39 | 38 | 51 | 71 | 46 | 60 | 51 | 48 | 82 | 60 | 86 | 56 |
| Uni-Walker w/o KIS | 61 | 60 | 51 | 55 | 43 | 46 | 56 | 74 | 48 | 68 | 59 | 49 | 86 | 63 | 86 | 60 |
| Uni-Walker w/o SSC | 60 | 59 | 49 | 56 | 42 | 45 | 56 | 75 | 48 | 69 | 58 | 47 | 84 | 62 | 86 | 60 |
| Uni-Walker w/o ECAS | 59 | 58 | 48 | 53 | 42 | 45 | 55 | 72 | 46 | 62 | 55 | 46 | 82 | 62 | 86 | 58 |
| **Uni-Walker (ours)** | **67** | **67** | **75** | **67** | **57** | **50** | **67** | **83** | **50** | **73** | **62** | **53** | **88** | **65** | **86** | **67** |

Table 14: Ablation study results (**F-SR** ↓, %) for the methods with task-shared navigation knowledge exploration.

| Methods | S1 | S2 | S3 | S4 | S5 | S6 | S7 | S8 | S9 | S10 | S11 | S12 | S13 | S14 | S15 | Avg. |
|---|---|---|---|---|---|---|---|---|---|---|---|---|---|---|---|---|
| Baseline | 29 | 20 | 42 | 36 | 35 | 27 | 27 | 16 | 12 | 20 | 22 | 14 | 9 | 8 | 0 | 21 |
| Uni-Walker w/o KIS | 13 | 14 | 35 | 31 | 28 | 12 | 20 | 13 | 8 | 9 | 9 | 13 | 4 | 3 | 0 | 14 |
| Uni-Walker w/o SSC | 14 | 16 | 38 | 30 | 30 | 13 | 20 | 12 | 8 | 8 | 11 | 16 | 7 | 5 | 0 | 15 |
| Uni-Walker w/o ECAS | 16 | 17 | 39 | 34 | 30 | 13 | 21 | 15 | 12 | 17 | 15 | 18 | 9 | 5 | 0 | 17 |
| **Uni-Walker (ours)** | **4** | **4** | **5** | **16** | **5** | **4** | **4** | **2** | **4** | **3** | **5** | **5** | **2** | **0** | **0** | **4** |

Table 15: Ablation study results (**SPL** ↑, %) for the methods with task-shared navigation knowledge exploration.

| Methods | S1 | S2 | S3 | S4 | S5 | S6 | S7 | S8 | S9 | S10 | S11 | S12 | S13 | S14 | S15 | Avg. |
|---|---|---|---|---|---|---|---|---|---|---|---|---|---|---|---|---|
| Baseline | 29 | 30 | 30 | 42 | 25 | 23 | 32 | 35 | 24 | 33 | 34 | 25 | 65 | 48 | 80 | 37 |
| Uni-Walker w/o KIS | 36 | 37 | 38 | 49 | 33 | 35 | 48 | **75** | **51** | 46 | 46 | 45 | 78 | 56 | 80 | 50 |
| Uni-Walker w/o SSC | 36 | 36 | 37 | 48 | 33 | 35 | 46 | 43 | 36 | 43 | 45 | 42 | 76 | 51 | 80 | 46 |
| Uni-Walker w/o ECAS | 35 | 36 | 35 | 49 | 32 | 32 | 45 | 43 | 35 | 42 | 42 | 40 | 75 | 49 | 80 | 45 |
| **Uni-Walker (ours)** | **66** | **63** | **59** | **61** | **49** | **50** | **65** | 74 | 50 | **72** | **56** | **51** | **82** | **56** | 80 | **62** |

the task-wise ablation results (SR) are summarized in Table 19, and the task-wise ablation results (F-SR) are summarized in Table 20, and the task-wise ablation results (SPL) are summarized in Table 21, and the task-wise ablation results (F-SPL) are summarized in Table 22, and the task-wise ablation results (OSR) are summarized in Table 23, and the task-wise ablation results (F-OSR) are summarized in Table 24.

# E  COMPARISON WITH OTHER EMBODIED NAVIGATION AGENTS

We construct universal embodied navigation agents through lifelong learning. We also perform comparison experiments with the other state-of-the-art universal navigation agents that are pre-trained

Table 16: Ablation study results (**F-SPL ↓**, %) for the methods with task-shared navigation knowledge exploration.

| Methods | S1 | S2 | S3 | S4 | S5 | S6 | S7 | S8 | S9 | S10 | S11 | S12 | S13 | S14 | S15 | Avg. |
|---|---|---|---|---|---|---|---|---|---|---|---|---|---|---|---|---|
| Baseline | 57 | 54 | 52 | 35 | 52 | 56 | 54 | 59 | 55 | 56 | 42 | 56 | 24 | 17 | 0 | 45 |
| Uni-Walker w/o KIS | 47 | 43 | 39 | 25 | 37 | 33 | 30 | 12 | 4 | 39 | 22 | 21 | 9 | 3 | 0 | 24 |
| Uni-Walker w/o SSC | 47 | 45 | 40 | 26 | 37 | 33 | 33 | 49 | 32 | 43 | 24 | 26 | 12 | 12 | 0 | 31 |
| Uni-Walker w/o ECAS | 49 | 45 | 44 | 25 | 38 | 38 | 35 | 49 | 34 | 44 | 29 | 30 | 13 | 16 | 0 | 32 |
| **Uni-Walker (ours)** | **3** | **3** | **5** | **6** | **6** | **4** | **6** | **13** | **6** | **4** | **5** | **11** | **5** | **3** | 0 | **6** |

Table 17: Ablation study results (**OSR ↑**, %) for the methods with task-shared navigation knowledge exploration.

| Methods | S1 | S2 | S3 | S4 | S5 | S6 | S7 | S8 | S9 | S10 | S11 | S12 | S13 | S14 | S15 | Avg. |
|---|---|---|---|---|---|---|---|---|---|---|---|---|---|---|---|---|
| Baseline | 75 | 80 | 79 | 63 | 81 | 81 | 70 | 82 | 62 | 70 | 61 | 80 | 91 | 75 | 100 | 77 |
| Uni-Walker w/o KIS | 77 | 82 | 78 | 65 | 82 | 88 | 71 | 82 | 62 | 70 | 61 | 80 | 91 | 75 | 100 | 78 |
| Uni-Walker w/o SSC | 78 | 81 | 79 | 66 | 82 | 89 | 71 | 82 | 61 | 71 | 62 | 80 | 91 | 75 | 100 | 78 |
| Uni-Walker w/o ECAS | 77 | 81 | 79 | 66 | 81 | 88 | 72 | 82 | 62 | 71 | 61 | 81 | 95 | 78 | 100 | 78 |
| **Uni-Walker (ours)** | **82** | **87** | **88** | **67** | **90** | **90** | **72** | **83** | **63** | **73** | **62** | **84** | **100** | **78** | 100 | **81** |

Table 18: Ablation study results (**OSR-F ↓**, %) for the methods with task-shared navigation knowledge exploration.

| Methods | S1 | S2 | S3 | S4 | S5 | S6 | S7 | S8 | S9 | S10 | S11 | S12 | S13 | S14 | S15 | Avg. |
|---|---|---|---|---|---|---|---|---|---|---|---|---|---|---|---|---|
| Baseline | 12 | 10 | 12 | 10 | 12 | 12 | 10 | 4 | 9 | 7 | 6 | 7 | 9 | 12 | 0 | 9 |
| Uni-Walker w/o KIS | 9 | 8 | 13 | 7 | 11 | 4 | 9 | 4 | 9 | 7 | 6 | 7 | 9 | 12 | 0 | 8 |
| Uni-Walker w/o SSC | 8 | 9 | 12 | 6 | 11 | 3 | 9 | 4 | 10 | 5 | 5 | 7 | 9 | 12 | 0 | 7 |
| Uni-Walker w/o ECAS | 9 | 9 | 12 | 6 | 12 | 4 | 8 | 4 | 9 | 5 | 6 | 6 | 5 | 8 | 0 | 7 |
| **Uni-Walker (ours)** | **4** | **2** | **2** | **4** | **2** | **2** | **8** | **2** | **7** | **3** | **5** | **2** | **0** | **8** | **0** | **3** |

Table 19: Ablation study results (**SR ↑**, %) for the methods with task-specific navigation knowledge exploration.

| Methods | S1 | S2 | S3 | S4 | S5 | S6 | S7 | S8 | S9 | S10 | S11 | S12 | S13 | S14 | S15 | Avg. |
|---|---|---|---|---|---|---|---|---|---|---|---|---|---|---|---|---|
| Baseline | 45 | 50 | 42 | 46 | 42 | 32 | 42 | 60 | 41 | 52 | 48 | 41 | 76 | 59 | 71 | 50 |
| Uni-Walker w/o ESOC | 65 | 62 | 70 | 61 | 54 | 45 | 59 | 78 | 48 | 69 | 59 | 50 | 85 | 63 | 85 | 64 |
| Uni-Walker w/o NSCoT | 46 | 51 | 43 | 46 | 43 | 33 | 45 | 61 | 42 | 55 | 49 | 44 | 77 | 60 | 71 | 51 |
| **Uni-Walker (ours)** | **67** | **67** | **75** | **67** | **57** | **50** | **67** | **83** | **50** | **73** | **62** | **53** | **88** | **65** | **86** | **67** |

Table 20: Ablation study results (**F-SR ↓**, %) for the methods with task-specific navigation knowledge exploration.

| Methods | S1 | S2 | S3 | S4 | S5 | S6 | S7 | S8 | S9 | S10 | S11 | S12 | S13 | S14 | S15 | Avg. |
|---|---|---|---|---|---|---|---|---|---|---|---|---|---|---|---|---|
| Baseline | 36 | 29 | 47 | 43 | 30 | 38 | 40 | 29 | 21 | 31 | 26 | 27 | 16 | 9 | 17 | 29 |
| Uni-Walker w/o ESOC | 7 | 11 | 11 | 24 | 10 | 13 | 16 | 8 | 8 | 8 | 9 | 11 | 6 | 3 | 1 | 10 |
| Uni-Walker w/o NSCoT | 34 | 27 | 46 | 43 | 28 | 37 | 36 | 28 | 19 | 27 | 25 | 21 | 14 | 8 | 17 | 27 |
| **Uni-Walker (ours)** | **4** | **4** | **5** | **16** | **5** | **4** | **4** | **2** | **4** | **3** | **5** | **5** | **2** | **0** | **0** | **4** |

on large-scale datasets. The results are summarized in Table 25. Based on the results, although these agents undergo large-scale joint training across multiple navigation tasks, they struggle to generalize across all tasks and often lack continual adaptability to ever-changing new navigation scenarios. Uni-

Table 21: Ablation study results (**SPL** ↑, %) for the methods with task-specific navigation knowledge exploration.

| Methods | S1 | S2 | S3 | S4 | S5 | S6 | S7 | S8 | S9 | S10 | S11 | S12 | S13 | S14 | S15 | Avg. |
|---|---|---|---|---|---|---|---|---|---|---|---|---|---|---|---|---|
| Baseline | 27 | 28 | 29 | 41 | 22 | 20 | 30 | 31 | 22 | 31 | 30 | 22 | 60 | 40 | 75 | 34 |
| Uni-Walker w/o ESOC | 65 | 61 | 55 | 60 | 45 | 49 | 64 | 73 | 48 | 70 | 55 | 50 | 81 | 55 | 78 | 61 |
| Uni-Walker w/o NSCoT | 29 | 31 | 31 | 42 | 23 | 21 | 32 | 33 | 25 | 33 | 31 | 23 | 61 | 41 | 76 | 35 |
| **Uni-Walker (ours)** | **66** | **63** | **59** | **61** | **49** | **50** | **65** | **74** | **50** | **72** | **56** | **51** | **82** | **56** | **80** | **62** |

Table 22: Ablation study results (**F-SPL** ↓, %) for the methods with task-specific navigation knowledge exploration.

| Methods | S1 | S2 | S3 | S4 | S5 | S6 | S7 | S8 | S9 | S10 | S11 | S12 | S13 | S14 | S15 | Avg. |
|---|---|---|---|---|---|---|---|---|---|---|---|---|---|---|---|---|
| Baseline | 60 | 57 | 53 | 37 | 58 | 62 | 57 | 64 | 58 | 59 | 49 | 61 | 30 | 31 | 6 | 45 |
| Uni-Walker w/o ESOC | 4 | 6 | 11 | 8 | 13 | 6 | 7 | 14 | 9 | 7 | 7 | 12 | 6 | 5 | 3 | 8 |
| Uni-Walker w/o NSCoT | 57 | 52 | 50 | 35 | 56 | 60 | 54 | 61 | 53 | 56 | 47 | 60 | 29 | 29 | 5 | 46 |
| **Uni-Walker (ours)** | **3** | **3** | **5** | **6** | **6** | **4** | **6** | **13** | **6** | **4** | **5** | **11** | **5** | **3** | **0** | **6** |

Table 23: Ablation study results (**OSR** ↑, %) for the methods with task-specific navigation knowledge exploration.

| Methods | S1 | S2 | S3 | S4 | S5 | S6 | S7 | S8 | S9 | S10 | S11 | S12 | S13 | S14 | S15 | Avg. |
|---|---|---|---|---|---|---|---|---|---|---|---|---|---|---|---|---|
| Baseline | 65 | 71 | 70 | 53 | 77 | 80 | 66 | 81 | 60 | 70 | 59 | 77 | 85 | 71 | 100 | 72 |
| Uni-Walker w/o ESOC | 80 | 86 | 79 | 66 | 89 | 89 | 71 | 82 | 63 | 70 | 62 | 84 | 100 | 75 | 100 | 80 |
| Uni-Walker w/o NSCoT | 67 | 73 | 72 | 55 | 80 | 82 | 67 | 82 | 62 | 70 | 61 | 81 | 100 | 78 | 100 | 75 |
| **Uni-Walker (ours)** | **82** | **87** | **88** | **67** | **90** | **90** | **72** | **83** | **63** | **73** | **62** | **84** | **100** | **78** | **100** | **81** |

Table 24: Ablation study results (**F-OSR** ↓, %) for the methods with task-specific navigation knowledge exploration.

| Methods | S1 | S2 | S3 | S4 | S5 | S6 | S7 | S8 | S9 | S10 | S11 | S12 | S13 | S14 | S15 | Avg. |
|---|---|---|---|---|---|---|---|---|---|---|---|---|---|---|---|---|
| Baseline | 24 | 20 | 22 | 24 | 16 | 13 | 15 | 5 | 12 | 7 | 9 | 10 | 15 | 16 | 0 | 14 |
| Uni-Walker w/o ESOC | 6 | 3 | 12 | 6 | 3 | 3 | 9 | 4 | 7 | 7 | 5 | 2 | 0 | 12 | 0 | 5 |
| Uni-Walker w/o NSCoT | 21 | 18 | 20 | 21 | 13 | 11 | 14 | 4 | 9 | 7 | 6 | 6 | 0 | 8 | 0 | 11 |
| **Uni-Walker (ours)** | **4** | **2** | **2** | **4** | **2** | **2** | **8** | **2** | **7** | **3** | **5** | **2** | **0** | **8** | **0** | **3** |

Walker demonstrates the feasibility of constructing a universal embodied navigation agent through lifelong learning.

# F  DISCUSSIONS ON THE PROPOSED COMPONENTS

As shown in our ablation studies (Tables 4–6), every component of Uni-Walker provides performance improvement. Below, we summarize their roles and contributions.

**Knowledge Inheritance Strategy (KIS).** KIS equips Uni-Walker with a human-inspired capability to efficiently acquire new skills by leveraging previously learned shared knowledge. It initializes the new expert subspace using relevant past knowledge (Eq. 2), enabling efficient adaptation. Removing KIS results in a 7.0% SR degradation.

**Experts Co-Activation Strategy (ECAS).** ECAS is the core architectural mechanism of DE-LoRA: it directly activates previously learned experts that are most relevant to the current task (Eq. 3), enabling knowledge sharing across tasks. Without ECAS, SR decreases by 9.2

Table 25: Comparison of the other state-of-the-art universal navigation agents: averaged metrics on the 18-task with our LENL settings.

| Comparison Methods | Avg SR ↑ (%) | Avg SPL ↑ (%) | Avg OSR ↑ (%) |
|---|---|---|---|
| NaviLLM Zheng et al. (2024) | 50 | 39 | 58 |
| ScaleVLN Wang et al. (2023b) | 53 | 42 | 61 |
| SAME Zhou et al. (2024) | 55 | 45 | 62 |
| Uni-Walker | **66** | **61** | **81** |

**Shared Smoothing Consolidation (SSC).** SSC is in order to progressively refine the shared subspace $A$, mitigating catastrophic forgetting of previous tasks (Eq. 4). Removing SSC leads to a 7.6% SR degradation.

**Expert Subspace Orthogonality Constraint (ESOC).** ESOC performs orthogonal constraint across expert subspaces $B$, promoting decoupled representation of navigation knowledge and preventing subspace mixing (Eq. 7). Its removal causes 3.8% SR degradation.

**Navigation-Specific Chain of Thought (NSCoT).** NSCoT provides task-style–specific reasoning for different instruction styles. It invokes relevant pretrained LLM knowledge and guides the agent toward more targeted decision-reasoning. Removing NSCoT yields the largest drop (16.2% SR), showing that style-aware reasoning is crucial for LLM-based navigation

**Task-Aware Knowledge Aggregation (TAKA).** TAKA is indispensable for task-ID–agnostic inference. It selects the most relevant experts using mixed matching and enables Uni-Walker to perform navigation in ID-agnostic task. We also provide ablation results that validate the effectiveness of the mixed matching strategy used by TAKA.

In summary, ECAS, TAKA, and NSCoT are the core components for accomplishing the LENL task, while the training strategies KIS, SSC, and ESOC promote disentangled representation of embodied navigation knowledge and further improve performance. We suggest that since ESOC has a relatively low impact (only +3.8%), it can be selectively removed when users have to simplify the Uni-Walker architecture.

# G DISCUSSIONS AND FUTURE WORKS

## FUTURE WORKS

While Uni-Walker demonstrates improved lifelong navigation performance under the LENL protocol, several future works remain: **Sim-to-real gap.** Experiments are conducted in simulated Matterport3D environments; transferring learned behaviors to physical robots will introduce perception, dynamics, and sensor-noise challenges that the current framework does not explicitly address. **Broader embodied task generalization.** Extend high-order adaptation principles to other embodied tasks, such as object manipulation, multi-agent coordination, or language-guided manipulation, and evaluate whether the DE-LoRA + TAKA paradigm generalizes beyond navigation.

## SOCIETAL IMPACTS

Our work aims to improve the adaptability and longevity of embodied agents, which can enable beneficial applications in assistive robotics, inspection, and disaster response. At the same time, several societal risks should be considered: **Privacy and surveillance.** Navigation agents operating in private or sensitive spaces may collect and store visual and spatial data. Mitigation: adopt strict data governance, on-device processing, and data minimization; use privacy-preserving techniques (e.g., differential privacy, anonymization) where appropriate. **Safety and misuse.** Autonomous navigation systems may cause harm if they behave unpredictably in crowded or safety-critical settings. Mitigation: incorporate verifiable safety constraints, human-in-the-loop control, conservative fallback behaviors, and thorough scenario-based testing before deployment.

