# OpenReview forum: "Lifelong Embodied Navigation Learning"
_ICLR.cc/2026/Conference — ICLR 2026 Poster_

### Official Review · Reviewer_Fhsy · 2025-10-16

**Soundness:** 3
**Presentation:** 3
**Contribution:** 3
**Rating:** 6
**Confidence:** 4

**Summary:**

This paper addresses the challenge of lifelong embodied navigation (LENL), where an agent must continually adapt to new navigation tasks without catastrophic forgetting. The authors propose Uni-Walker, a novel framework that combines LoRA-based expert decomposition with task-aware knowledge aggregation (TAKA) to effectively manage task-specific and shared knowledge across sequential navigation tasks. Uni-Walker is evaluated on a newly introduced LENL benchmark, comprising 18 task sequences with varying instruction styles and environments. Results show significant improvements over state-of-the-art continual learning methods, achieving an average SPL of 61% compared to 38% for the strongest baseline (SD-LoRA+TAKA). The work highlights the importance of efficient knowledge reuse and dynamic adaptation in lifelong learning scenarios.

**Strengths:**

- **Originality**:  The paper tackles a timely and important problem—how to enable agents to continuously learn new navigation tasks while avoiding catastrophic forgetting. The proposed Uni-Walker framework introduces a novel combination of LoRA experts and CLIP-based retrieval for task-aware knowledge aggregation, offering a fresh perspective on how to balance shared and task-specific knowledge in lifelong learning settings.

- **Quality**: The experimental design is rigorous and comprehensive. The authors introduce a new LENL benchmark consisting of 18 task sequences, covering various instruction styles and environments. They report detailed results and demonstrate consistent performance improvements across early and late tasks. Extensive ablation studies validate the effectiveness of each component (DE-LoRA, TAKA) and provide insights into their contributions.

- **Clarity**:  The paper is well-written and clearly structured. Key concepts are explained with clear diagrams and pseudocode. The overall framework is well illustrated and the evaluation setup helps readers understand the technical details. The comparison with prior work is fair and balanced, acknowledging both the strengths and limitations of existing approaches.

**Weaknesses:**

- **Limited evaluation**: While the experiments show strong performance on the LENL benchmark, it would be beneficial to evaluate Uni-Walker on additional datasets or more diverse environments and benchmarks to assess its generalizability. The current focus on indoor navigation tasks may limit the broader applicability of the findings.

- **Insufficient comparison with non-LoRA methods**: The paper primarily compares Uni-Walker with other LoRA-based continual learning methods (e.g., MoLA, BranchLoRA). Including comparisons with other continual learning methods would provide a more comprehensive understanding of Uni-Walker’s relative advantages and limitations.

- **Real World Applications**: The success of TAKA depends heavily on the quality of CLIP embeddings for aligning instructions and observations. If there is significant semantic drift or if instructions vary widely in style, the retrieval mechanism might degrade. Furthermore, the paper emphasizes “training-free” inference, but it still requires training a new expert for each incoming task. Clarifying the assumptions about task boundaries and the feasibility of applying Uni-Walker in truly online settings would strengthen the claims.

**Questions:**

See Weaknesses

---

> ### Author Response · Authors · 2025-11-20
> **Response to Reviewer Fhsy (1 / 2)**
>
> **1. Additional more diverse experiments (real world). (Weaknesses-1)**
>
> Thank you for the helpful suggestion. Following your suggestion, we add real-world deployment experiments to further assess the generalizability of Uni-Walker beyond indoor simulation environments. To enable a 7B LLM–based agent to operate on real robots, we designed a sensing - WiFi transmission - server-side computation pipeline. Uni-Walker is deployed on a navigation platform built upon DeepRobotDog Lite2, equipped with robust onboard multiple cameras (Hikvision DS-E12) to mitigate sensing noise. The full system design is provided in the revised Appendix G. Beyond the original 18 simulation tasks, we additionally collect two real-world lifelong navigation tasks (one indoor and one outdoor). For each environment, we recorded 400 trajectories for training and performed 100 evaluation trials:
>
> |  Real-World Task  | Uni-Walker SR | SD-LoRA SR |
> | :---------------: | :----------: | :----------: |
> |  S19 (Indoor) |    58%  |    49%  |
> |  S20 (Outdoor) |   52%   |    44%  |
>
> **For examples of navigation visualization in practical deployments, please refer to the revised Appendix Figures 8-10.** These preliminary results demonstrate that Uni-Walker can effectively transfer to real-world robotic deployment, further supporting the scalability and practical utility of our framework in diverse environments.
>
> **2. More comparison with non-LoRA continual-learning baselines. (Weaknesses-2)**
>
> Thank you for your suggestion. i) We clarify why the initial version of the paper did not include non-LoRA Continual Learning (CL) baselines. Non-LoRA continual-learning methods (e.g., EWC, GEM, DER, iCaRL) require full-parameter updates of the backbone model. However, in our setting, the backbone is a 7B-scale LLM, and full-model fine-tuning is computationally prohibitive in both memory and optimization cost. This makes such methods infeasible for LLM-based embodied navigation, which is why prior large-model CL works also adopt LoRA-style adapters as the default paradigm. ii) To further strengthen the comparison, we additionally include two non-LoRA continual learning approaches: SEMA [1], a multi-expert adapter expansion CL method (CVPR 2025); NBAgent [2], a continual skill distillation framework for robotic manipulation (arXiv 2024). We evaluate these methods under the same LENL setting. The task-wise SR (%) results are shown below.
>
> | Methods                    | S1 | S2 | S3 | S4 | S5 | S6 | S7 | S8 | S9 | S10 | S11 | S12 | S13 | S14 | S15 | S16 | S17 | S18 | Avg. SR |
> |----------------------------|----|----|----|----|----|----|----|----|----|-----|-----|-----|-----|-----|-----|-----|-----|-----|-----|
> | Seq-FT                     | 0  | 0  | 0  | 5  | 4  | 0  | 5  | 18 | 8  | 9   | 8   | 21  | 20  | 25  | 86  | 0   | 0   | 0   | 12  |
> | LwF-LoRA                   | 0  | 5  | 0  | 5  | 6  | 0  | 5  | 22 | 8  | 12  | 10  | 30  | 45  | 36  | 85  | 5   | 0   | 0   | 15  |
> | EWC-LoRA                   | 0  | 5  | 0  | 10 | 4  | 0  | 5  | 20 | 8  | 9   | 8   | 31  | 49  | 40  | 86  | 5   | 5   | 0   | 16  |
> | Dense MoLE                 | 10 | 10 | 4  | 10 | 8  | 3  | 10 | 18 | 16 | 19  | 17  | 30  | 56  | 51  | 85  | 12  | 10  | 8   | 21  |
> | Sparse MoLE                | 15 | 12 | 0  | 17 | 6  | 6  | 16 | 18 | 18 | 24  | 15  | 32  | 58  | 53  | 84  | 16  | 12  | 10  | 23  |
> | MoLA                       | 15 | 12 | 0  | 12 | 6  | 10 | 20 | 18 | 13 | 26  | 19  | 34  | 61  | 59  | 86  | 20  | 12  | 11  | 24  |
> | HydraLoRA                  | 16 | 14 | 15 | 19 | 6  | 15 | 16 | 24 | 17 | 17  | 21  | 41  | 68  | 57  | 86  | 18  | 14  | 16  | 27  |
> | BranchLoRA                 | 26 | 20 | 20 | 25 | 8  | 21 | 20 | 26 | 20 | 18  | 22  | 39  | 78  | 54  | 86  | 28  | 20  | 15  | 30  |
> | O-LoRA + TAKA              | 55 | 61 | 50 | 54 | 42 | 42 | 54 | 79 | 48 | 66  | 56  | 54  | 88  | 63  | 86  | 65  | 53  | 36  | 58  |
> | SD-LoRA + TAKA             | 58 | 57 | 54 | 46 | 49 | 42 | 57 | 76 | 51 | 59  | 57  | 54  | 81  | 62  | 87  | 68  | 55  | 48  | 59  |
> | **SEMA** [1] | 38 | 34 | 32 | 34 | 19 | 32 | 33 | 38 | 42 | 30  | 33  | 51  |  84 | 61  | 86  | 39 | 42 | 36 | 43 |
> | **NBAgent** [2]   | 29 | 26 | 23 | 26 | 10 | 23 | 23 | 28 | 24 | 22  | 26  | 42  | 81  | 57  | 86  | 31  | 24  | 17  | 33 |
> | **Uni-Walker (ours)**      | 67 | 67 | 75 | 67 | 57 | 50 | 67 | 83 | 50 | 73  | 62  | 53  | 88  | 65  | 86  | 74  | 61  | 51  | 66  |
>
> Across all 18 tasks, Uni-Walker still achieves the highest average performance, outperforming both LoRA-based and non-LoRA continual learning methods.
>
> [1] Self-Expansion of Pre-trained Models with Mixture of Adapters for Continual Learning. CVPR 2025.
> [2] Never-Ending Behavior-Cloning Agent for Robotic Manipulation. arXiv 2024.

---

> ### Author Response · Authors · 2025-11-20
> **Response to Reviewer Fhsy (2 / 2)**
>
> **3. Discussion on retrieval mechanisms and clarification of task boundaries. (Weaknesses-3)**
>
> Thank you for your constructive comments. We provide further clarification regarding the robustness of TAKA and the task boundaries.
>
> **I) Robustness of TAKA Retrieval.** i) CLIP-based embedding alignment retrieval is the most commonly used method for visual-language modal alignment. Since the CLIP model has been trained with large-scale multimodal data, its feature extraction capability possesses a degree of robustness for semantic drift. We acknowledge that severe semantic drift leads to matching degradation, which remains an open challenge for the entire multimodal alignment community. ii）Our TAKA relies on relative similarity rather than absolute similarity scores. The greater the semantic differences between tasks, the larger the relative similarity gap, naturally improving retrieval performance. Even when the number of task diversities grows significantly, the extracted representations remain relatively highly discriminative. To validate large-scale and diversity scalability, we conduct an extended retrieval experiment across all 90 scenes in the Matterport3D Simulator. The retrieval success rate remains consistently high regardless of the number of tasks:
>
> | Metric                 | 10 Tasks | 20 Tasks | 30 Tasks | 40 Tasks | 50 Tasks | 60 Tasks | 70 Tasks | 80 Tasks | 90 Tasks |
> |:----------------------:|:--------:|:--------:|:--------:|:--------:|:--------:|:--------:|:--------:|:--------:|:--------:|
> | Retrieval SR |  99.8%   |  99.7%   |  99.8%   |  99.7%   |  99.8%   |  99.7%   |  99.8%   |  99.7%   |  99.8%   |
>
> These results clearly show that TAKA’s matching ability does not degrade as the number of task diversities grows, confirming its scalability and robustness.
>
> **II) Clarification on task boundaries.** i) During training, Uni-Walker introduces one lightweight expert per task (≈2.1 MB), allowing the model to efficiently acquire task-specific navigation skills. In real deployments, users may freely decide whether a new expert is needed for a new environment. Furthermore, Uni-Walker can automatically determine task boundaries by comparing TAKA's absolute match scores against a threshold. Specifically, if the highest expert similarity (Eq. 10) exceeds a threshold, the system infers that the current task is sufficiently similar to a previously learned task and no new expert is created; otherwise, a new expert is allocated. To evaluate the feasibility of this mechanism under true real-world online settings, we adopt a threshold of 0.95 and compare Uni-Walker with its Auto-Uni-Walker variant. The SR (%) results are summarized below:
>
> | Methods            | S1 | S2 | S3 | S4 | S5 | S6 | S7 | S8 | S9 | S10 | S11 | S12 | S13 | S14 | S15 | Avg. SR  |
> |--------------------|----|----|----|----|----|----|----|----|----|-----|-----|-----|-----|-----|-----|------|
> | Uni-Walker         | 67 | 67 | 75 | 67 | 57 | 50 | 67 | 83 | 50 | 73  | 62  | 53  | 88  | 65  | 86  | 67.33 |
> | Auto-Uni-Walker  | 67 | 68 | 74 | 67 | 58 | 51 | 67 | 82 | 50 | 72  | 63  | 53  | 87  | 64  | 86  | 67.26 |
>
> The results show no significant performance change, demonstrating that Uni-Walker remains stable and effective even when task boundaries are automatically determined. This supports the feasibility of applying Uni-Walker in real-world online lifelong navigation scenarios. ii) On the other hand, following the assumption in [1], if the LoRA parameter subspaces for each task are absolutely orthogonal, then the maximum number of supported tasks is bounded by [d/r], where d is the dimension of the hidden layer in the LLM, and r is the rank of the LoRA parameter space. Therefore, based on this assumption, theoretically, the maximum number of supported tasks is 4096/16 = 256. In practice, each of our LoRA expert subspaces is not strictly orthogonal and exhibits co-activation, resulting in a maximum number of tasks exceeding 256. iii) In addition, based on the theoretical experience provided in [2], with the CLIP latent space dimension of 1024, the maximum number of retrievable results is 4,031,487 (1024^3 × 0.0037 + 1024^2 × 0.0520 + 1024 × 4.0309 − 10.5322).
>
> [1] BiLoRA: Almost-orthogonal Parameter Spaces for Continual Learning. CVPR 2025.
>
> [2] On the Theoretical Limitations of Embedding-Based Retrieval. arXiv 2025.

---

> > ### Comment · Reviewer_Fhsy · 2025-11-27
> > **Revision answers to my concerns**
> >
> > Thank you for the response. I have also read other reviews and response. Since I've found no additional concern, I will keep my original rating.

---

> > > ### Author Response · Authors · 2025-11-27
> > > **Official Comment by Authors**
> > >
> > > Thank you for taking the time to review our work and our rebuttal. We truly appreciate your thoughtful comments and are glad that no additional concerns were raised. If any further clarification would be helpful, we would be more than happy to provide it.

---

### Official Review · Reviewer_AqL1 · 2025-10-30

**Soundness:** 3
**Presentation:** 3
**Contribution:** 3
**Rating:** 6
**Confidence:** 3

**Summary:**

In this work, the authors introduce a lifelong embodied navigation learning task, where an agent is required to continually acquire new navigation skills, without catastrophic forgetting on the previously learned knowledge. To achieve this goal, they design a Uni-Walker to decouple navigation knowledge into task-shared and task-specific components. Then they design different strategies to learn shared and specific knowledge for life-long learning.

**Strengths:**

* Novelty

In this work, the author propose a kind of new task in navigation, namely, lifelong embodied navigation learning. This task itself is interesting and practical for embodied navigation in the real world.

* Clarity

The paper is with good structure. Hence, the clarity is basically good.

* Significance

This paper focuses on a new task of lifelong embodied navigation learning. It would possibly bring new research works for navigation. Hence, the significance is basically OK for this community.

**Weaknesses:**

* Method

Even though the task is kind of new, the techniques to address this task are relatively straightforward. It basically leverages LORA adaption in the decoder for decoupling navigation knowledge into shared and specific parts. Then, they integrate the existing strategies to learn these two types of knowledge for life long learning. Please futher clarify the key novel design in the proposed framework, instead of adapting and integrating the straightforward strategies together on a newly-defined task.

* Figure Issue

 The figures (e.g., Fig 1-2) should be further re-arranged. The words are too small to see. The whole figure looks very crowded.

**Questions:**

Please see the weakness section.

---

> ### Author Response · Authors · 2025-11-20
> **Response to Reviewer AqL1**
>
> Thank you for your careful reading of our manuscript and for the constructive comments. We sincerely appreciate your recognition of the proposed Lifelong Embodied Navigation Learning task and the strengths of our Uni-Walker framework. Below, we address your concerns.
>
> **1. Please further clarify the key novel design in the proposed framework, instead of adapting and integrating the straightforward strategies together on a newly defined task. (Weaknesses-1)**
>
> Thank you for your constructive comments. Besides, our work introduces the new lifelong embodied navigation learning (LENL) task; the proposed framework is not a straightforward integration of existing LoRA-based techniques. Each component is specifically tailored for the unique challenges of LENL. **Different from the existing LoRA-based continual learning methods, we propose a Decoder Extension LoRA (DE-LoRA) architecture**, which explicitly decomposes navigation knowledge into a shared subspace A (capturing fundamental navigation skills) and multiple extension expert subspaces B (capturing scene-specific and instruction-specific knowledge). On top of DE-LoRA, **we design an Experts Co-Activation Strategy (ECAS),** which enables agents to directly activate the experts most relevant to the current task from previous tasks, with **multiple experts activated simultaneously** to achieve knowledge transfer and exploitation. The designed ECAS is critical for efficient new task adaptation and is substantially different from conventional LoRA, which learn each task independently. Furthermore, our KIS and ESOC training strategies further reinforce shared knowledge consolidation and specific knowledge decoupling, which effectively prevents catastrophic forgetting in the LENL setting. Finally, to approach realistic embodied navigation scenarios, we design a Task-Aware Knowledge Aggregation (TAKA) mechanism. **TAKA allows the agent to activate the most relevant experts without a task-ID during inference**, enabling generalization to unseen navigation tasks. In summary, besides our task being new, the proposed DE-LoRA with ECAS, KIS, ESOC, and TAKA are all purposefully designed for the LENL problem, rather than straightforward combinations of existing strategies.
>
> **2. The figures should be further re-arranged. (Weaknesses-2)**
>
> Thank you for pointing out the figure readability issue. We have revised all the figures in our paper. These revisions will substantially improve clarity and make the overall framework easier to follow.

---

> > ### Comment · Reviewer_AqL1 · 2025-11-28
> > **Feedback**
> >
> > Thanks for the response. The main concern is addressed. I keep my original rating.

---

> > > ### Author Response · Authors · 2025-11-28
> > > **Official Comment by Authors**
> > >
> > > We sincerely thank the reviewer for the response and for the recognition of our work. If the reviewer has any further questions, we would be very happy to discuss them.

---

### Official Review · Reviewer_5sW8 · 2025-10-31

**Soundness:** 3
**Presentation:** 3
**Contribution:** 3
**Rating:** 6
**Confidence:** 3

**Summary:**

This paper presents a comprehensive approach to Lifelong Embodied Navigation Learning (LENL), a novel problem setting where embodied agents must continually adapt to new navigation tasks across diverse scenes and instruction styles while mitigating catastrophic forgetting of previously acquired knowledge. The authors propose Uni-Walker, a framework built upon a Decoder Extension LoRA (DE-LoRA) architecture that explicitly decouples navigation knowledge into task-shared and task-specific components.

Uni-Walker introduces several key strategies:
- Knowledge Inheritance Strategy (KIS) and Experts Co-Activation Strategy (ECAS) to facilitate the transfer and refinement of task-shared knowledge.
- Expert Subspace Orthogonality Constraint (ESOC) and Navigation-Specific Chain-of-Thought (NSCoT) to capture and enhance task-specific knowledge and instruction understanding.
- Task-Aware Knowledge Aggregation (TAKA) for dynamically activating relevant experts during inference.

The paper also introduces a new lifelong embodied navigation benchmark consisting of 18 tasks with varying scenes and instruction styles (Vision-Language Navigation - VLN, Object Localization Navigation - OLN, Dialogue Understanding Navigation - DUN). Extensive experiments demonstrate that Uni-Walker significantly outperforms state-of-the-art LoRA-based continual learning methods and other universal navigation agents, achieving superior success rates and substantially lower forgetting rates across multiple metrics.

**Strengths:**

1. The paper introduces a novel problem (LENL) and a corresponding benchmark, which is a vital contribution. The Uni-Walker framework, with its DE-LoRA and specialized strategies (KIS, ECAS, ESOC, NSCoT, TAKA), demonstrates high originality in combining and extending existing ideas for the specific challenges of embodied lifelong learning.
2. The empirical results are robust and convincing. Uni-Walker achieves state-of-the-art performance, significantly reducing catastrophic forgetting compared to numerous strong baselines. The comprehensive ablation studies rigorously validate the effectiveness of each component, reinforcing the quality of the proposed solution.

**Weaknesses:**

1. While each component of Uni-Walker (DE-LoRA, KIS, ECAS, SSC, ESOC, NSCoT, TAKA) is well-justified, the overall framework is quite intricate with multiple interacting mechanisms. It might be challenging to disentangle the precise individual contributions or to simplify the architecture without performance degradation. A discussion on the trade-offs between complexity and performance, or potential avenues for simplification, could be beneficial.
2. While the new benchmark is excellent, 18 tasks, even with diverse scenes and instruction styles, might still be considered limited for truly "lifelong" learning in the vast and unpredictable real world. All scenes are from the Matterport3D simulator. A discussion on how the framework might scale to hundreds or thousands of tasks, or transfer to real-world robots (as briefly mentioned in future work), would strengthen the paper.
3. The paper lists several hyperparameters. While values are provided, a more detailed sensitivity analysis of the most critical hyperparameters could provide further insights into the robustness and general applicability of Uni-Walker.
4. The method involves managing multiple experts, computing Fisher Information Matrices, and performing retrieval for Task-Aware Knowledge Aggregation. While LoRA is parameter-efficient, the runtime and memory overhead for a very large number of tasks (e.g., hundreds or thousands) could become substantial. A more explicit discussion of the computational scalability with increasing task numbers would be valuable.
5. The paper mentions 3 tasks (S16, S17, S18) for unseen generalization. While these are included in the average results, a dedicated discussion or a separate table comparing Uni-Walker's performance specifically on these unseen tasks against baselines would highlight its true generalization capabilities more clearly.

**Questions:**

1. Could the authors elaborate on the computational and memory overhead of managing the Fisher Information Matrix (FA,t) and the retrieval index (Re) as the number of tasks (T) grows very large (e.g., hundreds or thousands)? Are there strategies to mitigate this growth, or is it a known limitation for extremely long lifelong learning sequences?
2. For the Task-Aware Knowledge Aggregation (TAKA), what is the sensitivity of Uni-Walker's performance to the choice of K (number of activated experts) and µ (similarity threshold for masking)? Have you experimented with adaptive ways to determine K or µ?
3. The paper states that "task-id t is agnostic during the testing phase." How is the instruction style (VLN, OLN, DUN) determined at inference time to apply the correct Navigation-Specific Chain-of-Thought (NSCoT)? Is it inferred from the user instruction, or is the instruction format implicitly indicating the style?
4. For the Knowledge Inheritance Strategy (KIS), PCA is applied to learned experts having the "same instruction knowledge." How is this grouping of "same instruction knowledge" determined? Is it hardcoded based on the benchmark's predefined instruction styles, or is there an automatic mechanism to identify similar instruction knowledge for new, previously unseen instruction styles?
5. Could the authors provide a more focused analysis or a separate table comparing the performance of Uni-Walker and the top-performing baselines specifically on the 3 "unseen generalization" tasks (S16, S17, S18)? This would provide clearer evidence of the framework's ability to generalize to truly novel environments/instruction combinations.

---

> ### Author Response · Authors · 2025-11-20
> **Response to Reviewer 5sW8 (1 / 4)**
>
> We sincerely thank you for the detailed evaluation and constructive feedback. Your acknowledgment of the novelty of the LENL problem and the effectiveness of Uni-Walker is greatly appreciated. Below, we address all the concerns.
>
> **1. Each component's contributions and the trade-offs between complexity and performance. (Weaknesses-1)**
>
> Thanks for your insightful comment. As shown in our ablation studies (Tables 4–6), every component of Uni-Walker provides performance improvement. Below, we summarize their roles and contributions.
>
> **Knowledge Inheritance Strategy (KIS).** KIS equips Uni-Walker with a human-inspired capability to efficiently acquire new skills by leveraging previously learned shared knowledge. It initializes the new expert subspace using relevant past knowledge (Eq. 2), enabling efficient adaptation. Removing KIS results in a 7.0% SR degradation, and training time decreases by 0.5 hours.
>
> **Experts Co-Activation Strategy (ECAS).** ECAS is the core architectural mechanism of DE-LoRA: it directly activates previously learned experts that are most relevant to the current task (Eq. 3), enabling knowledge sharing across tasks. Without ECAS, SR decreases by 9.2%, and training time decreases by 0.9 hours.
>
> **Shared Smoothing Consolidation (SSC).** SSC is in order to progressively refine the shared subspace $A$, mitigating catastrophic forgetting of previous tasks (Eq. 4). Removing SSC leads to a 7.6% SR degradation, and training time decreases by 0.4 hours.
>
> **Expert Subspace Orthogonality Constraint (ESOC).** ESOC performs orthogonal constraint across expert subspaces $B$, promoting decoupled representation of navigation knowledge and preventing subspace mixing (Eq. 7). Its removal causes 3.8% SR degradation, and training time decreases by 0.7 hours.
>
> **Navigation-Specific Chain of Thought (NSCoT).** NSCoT provides task-style–specific reasoning for different instruction styles. It invokes relevant pretrained LLM knowledge and guides the agent toward more targeted decision-reasoning. Removing NSCoT yields the largest drop (16.2% SR), training time decreases by 1.3 hours, showing that style-aware reasoning is crucial for LLM-based navigation.
>
> **Task-Aware Knowledge Aggregation (TAKA).** TAKA is indispensable for task-ID–agnostic inference. It selects the most relevant experts using mixed matching and enables Uni-Walker to perform navigation in an ID-agnostic task. We also provide ablation (Table 6) that validates the effectiveness of the mixed matching strategy used by TAKA.
>
> Although our Uni-Walker appears complex due to incorporating multiple strategies, the role of each strategy is clearly determined. Furthermore, the computational consumption added by these strategies (about a few M) is negligible compared to that of the LLM itself (7B). We suggest that, since ESOC has a relatively low performance impact (only +3.8%) and increased training time (0.7 hours), it can be selectively removed when users have to simplify the Uni-Walker architecture for trade-offs between complexity and performance.

---

> ### Author Response · Authors · 2025-11-20
> **Response to Reviewer 5sW8 (2 / 4)**
>
> **2. A discussion of the computational scalability with increasing task numbers. (Weaknesses-2 & Weaknesses-4 & Questions-1)**
>
> Thank you for acknowledging that the current 18-task benchmark already covers a wide range of diverse scenes and instruction styles. Below, we further discuss the theoretical scalability of Uni-Walker to hundreds or thousands of tasks.
>
> I) We analyze the resource cost of LoRA and the Fisher Information Matrix in our DE-LoRA architecture. For each additional task, Uni-Walker adds: ≈ 2.1 MB for the LoRA expert subspace and maintains a Fisher Matrix of the same size. Thus, even when the number of tasks grows beyond 100, the total storage remains modest: 100 × 4.2 MB ≈ 0.4 GB. Such overhead is negligible for modern LLM-based navigation systems (7B, 13B, or larger), demonstrating the practical scalability of the architecture.
>
> II) The TAKA maintains strong scalability even when the number of tasks grows to a very large scale. TAKA performs embedding-level retrieval using the large-scale pretrained CLIP model, and activates only the Top-K experts based on embedding similarity (Eq. 10). Since the matching process relies on **relative similarity rather than absolute similarity scores**, the retrieval quality does not degrade even as the total number of experts increases. i) To validate this, we conduct an extended retrieval experiment across all 90 scenes in the Matterport3D Simulator. For each scene, the agent randomly navigates 100 times and performs TAKA matching. The results show that the retrieval success rate remains consistently high and stable regardless of the number of tasks:
>
> | Metric                 | 10 Tasks | 20 Tasks | 30 Tasks | 40 Tasks | 50 Tasks | 60 Tasks | 70 Tasks | 80 Tasks | 90 Tasks |
> |:----------------------:|:--------:|:--------:|:--------:|:--------:|:--------:|:--------:|:--------:|:--------:|:--------:|
> | Retrieval SR |  99.8%   |  99.7%   |  99.8%   |  99.7%   |  99.8%   |  99.7%   |  99.8%   |  99.7%   |  99.8%   |
>
> These results demonstrate that TAKA's matching robustness does not diminish as the number of experts grows, confirming its practical scalability. ii) To further evaluate scalability under extremely large task pools, we conduct an additional scalability test. Specifically, after training on the first 10 tasks, we artificially add 1000 randomly initialized experts with their corresponding 1000 retrieval embeddings extracted from random natural scenarios by CLIP, simulating a very large-scale lifelong learning scenario. We then continue training Uni-Walker on the next 5 tasks and measure performance on tasks S10–S15. The SR (%) results are summarized below:
>
> | Methods           | S10 | S11 | S12 | S13 | S14 | S15 |  Avg. SR   |
> |-------------------|-----|-----|-----|-----|-----|-----|--------|
> | Uni-Walker        | 73  | 62  | 53  | 88  | 65  | 86  | 71.2  |
> | 1000+ Uni-Walker  | 74  | 61  | 54  | 88  | 64  | 86  | 71.2  |
>
> The results show no degradation when Uni-Walker operates alongside 1,000 additional experts. This confirms that both training stability and TAKA retrieval remain robust under very large expert pools. iii) In addition, based on the theoretical experience provided in [1], with the CLIP latent space dimension of 1024, the maximum number of retrievable results is 4,031,487 (1024^3 × 0.0037 + 1024^2 × 0.0520 + 1024 × 4.0309 − 10.5322).
>
> [1] On the Theoretical Limitations of Embedding-Based Retrieval. arXiv 2025.
>
> These results confirm that TAKA’s matching robustness does not degrade as the number of experts increases, and its storage cost (only 150 KB visual + 3.1 KB instruction embedding per task) remains lightweight.
>
> **3. Extension to real-world robotic experiments. (Weaknesses-2)**
>
> To address the deployment of a 7B LLM in real robots, we design a sensing–WiFi transmission–server computation pipeline. Specifically, we deploy Uni-Walker on a navigation platform built on DeepRobotDog Lite2, and multiple robust onboard cameras (Hikvision DS-E12) are used to reduce sensing noise. **The deployment system design is included in the revised Appendix G.** Beyond the 18 existing tasks, we additionally collected two real-world lifelong navigation tasks (one indoor, one outdoor). For each setting, we collect 400 navigation trajectories for training, and execute 100 evaluation trials per scene. The results are summarized below:
>
> |  Real-World Task  | Uni-Walker SR | SD-LoRA SR |
> | :---------------: | :----------: | :----------: |
> |  S19 (Indoor) |    58%  |    49%  |
> |  S20 (Outdoor) |   52%   |    44%  |
>
> **For examples of navigation visualization in practical deployments, please refer to the revised Appendix Figures 8-10.** These preliminary results demonstrate that Uni-Walker can successfully transfer to real-world robotic deployment, further supporting the scalability and practical utility of our framework.

---

> ### Author Response · Authors · 2025-11-20
> **Response to Reviewer 5sW8 (3 / 4)**
>
> **4. Sensitivity analysis of the most critical hyperparameters. (Weaknesses-3 & Questions-2)**
>
> Thanks for your invaluable comment. Following your suggestion, we conduct additional experiments to analyze the sensitivity of Uni-Walker to the hyperparameters K (the number of activated experts) and μ (the similarity threshold for masking).
>
> I) Sensitivity to the Number of Activated Experts K. We evaluate Uni-Walker with K = 1, 2, 3, 4. The average SR (%) across all lifelong tasks is summarized as follows:
>
> | Metric   |   K=1   |   K=2   |   K=3   |   K=4   |
> |:--------:|:-------:|:-------:|:-------:|:-------:|
> | Avg. SR  |  59.2   |  66.0   |  67.3   |  67.3   |
>
> The results show that setting K = 2 already achieves a near-optimal trade-off between performance and computational complexity. Increasing K beyond 2 yields only marginal improvements (≤1.3%), while K = 1 significantly underperforms. Therefore, for efficiency and stability, we choose K = 2 as our default value.
>
> II) Sensitivity to the Similarity Threshold μ. We evaluate μ from 0.1 to 0.9 in increments of 0.1. The resulting average SR (%) is shown below:
>
> | μ    | 0.1 | 0.2 | 0.3 | 0.4 | 0.5 | 0.6 | 0.7 | 0.8 | 0.9 |
> |:----:|:---:|:---:|:---:|:---:|:---:|:---:|:---:|:---:|:---:|
> | Avg. SR   |65.1 |65.1 |65.1 |65.8 |66.0 |66.0 |65.7 |28.5 | 0   |
>
> The results show that the performance remains stable in the range μ = 0.4–0.7, with SR around 65.7–66.0. Very low μ values behave similarly to having no masking and provide no benefit. Very high μ values mask out nearly all experts, leading to performance collapse (μ ≥ 0.8).
>
> Following your comment, we also explore an adaptive strategy to determine the masking threshold μ instead of using a fixed global value. Specifically, for each query instruction, we compute the similarity vector s against all expert retrieval embeddings and derive μ from the statistical properties of this distribution. We test three adaptive variants: μ = mean(s); μ = median(s); μ = mean(s) + 0.8 × std(s). This design allows μ to naturally adjust according to the distributional characteristics of the current query, making the matching process more robust. The average SR (%) on the LENL benchmark is shown below:
>
> | μ Strategy                     | Avg. SR |
> |:-----------------------------:|:--------:|
> | mean(s)                       |  65.1    |
> | median(s)                     |  65.1    |
> | mean(s) + 0.8 × std(s)        |  65.8    |
>
> These results indicate that the adaptive mechanisms achieve performance comparable to the fixed-threshold approach, while offering improved robustness to variation in query similarity distributions. We recommend using this adaptive μ in open-world scenarios with a larger number of tasks, where dynamic scaling becomes increasingly beneficial.
>
> **5. A dedicated discussion with a separate table for unseen generalization. (Weaknesses-5 & Questions-5)**
>
> Thank you for the valuable suggestion. Following your suggestion, we provide a dedicated comparison of the three unseen generalization tasks (S16, S17, S18). The SR (%) results are summarized below:
>
> | Method         |  S16 | S17 | S18 |  Avg. SR  |
> |:--------------:|:----:|:----:|:----:|:-----:|
> | HydraLoRA      |  18  |  14  |  16  | 16.0  |
> | BranchLoRA     |  28  |  20  |  15  | 21.0  |
> | O-LoRA + TAKA  |  65  |  53  |  36  | 51.3  |
> | SD-LoRA + TAKA |  68  |  55  |  48  | 57.0  |
> | **Uni-Walker** | **74** | **61** | **51** | **62.0** |
>
> Uni-Walker achieves the best performance across all unseen tasks, outperforming the SOTA baseline (SD-LoRA + our TAKA) by 5.0% (62.0% → 57.0% Avg SR). This superior generalization capability results from: Decoupled knowledge learning via DE-LoRA, ESOC, and KIS, ensuring that shared navigation knowledge is preserved while task-specific expertise remains disentangled rather than entangled or overwritten; Dynamic knowledge aggregation through TAKA, which activates the most relevant experts for unseen scenes, enabling the agent to reuse transferable skill knowledge when encountering unseen scenes. Together, these mechanisms allow Uni-Walker to adapt to new tasks while retaining previously learned knowledge, leading to significantly better performance in unseen generalization.

---

> ### Author Response · Authors · 2025-11-20
> **Response to Reviewer 5sW8 (4 / 4)**
>
> **6. How is the instruction style (VLN, OLN, DUN) determined at inference time to apply the correct NSCoT? (Questions-3)**
>
> Thank you for the comment. We clarify how the appropriate NSCoT is selected during inference. NSCoT provides three distinct reasoning processes tailored for the three navigation instruction styles (VLN, OLN, DUN), enabling the LLM to better activate its pretrained knowledge for the corresponding navigation task. In the testing phase, the instruction itself implicitly indicates the style. Specifically, the instruction naturally contains style-specific linguistic patterns: VLN: long, descriptive language sequences; OLN: goal-oriented patterns such as “find the X”; DUN: dialogue-like question–answer structures. Given these clear differences in linguistic structure, the LLM can reliably distinguish the instruction style and select the corresponding NSCoT template. Thus, the instruction style does not require any external task-ID; it is implicitly encoded in the instruction itself and effortlessly identified by the LLM.
>
> **7. How is this grouping of "same instruction knowledge" determined?  (Questions-4)**
>
> A7: Thank you for the comment. We clarify how “same instruction knowledge” is identified in the KIS. During training, the instruction styles are known. Thus, in our current LENL benchmark, the grouping is hardcoded based on the predefined three instruction styles (VLN, OLN, DUN). For new task learning, we extract the principal components from the corresponding previously learned experts with the same instruction style and use them to initialize the new expert, enabling efficient inheritance of shared knowledge. Moreover, even without predefined styles, determining the appropriate group remains straightforward. The input instruction itself contains a clear linguistic structure. Leveraging the LLM’s inherent language understanding capabilities, the instruction style can be reliably classified from the input instruction itself. Hence, automatically assigning new tasks to the correct knowledge group is essentially a simple classification problem and does not pose additional challenges. We also add a set of experiments that employ LLM to automatically determine the style of input instruction. The expert grouping labels are stored after training to allow for automatic grouping and KIS. The results are summarized below. The SR (%) results show no significant performance changes.
>
> | Methods            | S1 | S2 | S3 | S4 | S5 | S6 | S7 | S8 | S9 | S10 | S11 | S12 | S13 | S14 | S15 | Avg. SR  |
> |--------------------|----|----|----|----|----|----|----|----|----|-----|-----|-----|-----|-----|-----|------|
> | Uni-Walker         | 67 | 67 | 75 | 67 | 57 | 50 | 67 | 83 | 50 | 73  | 62  | 53  | 88  | 65  | 86  | 67.33 |
> | AutoKIS Uni-Walker  | 68 | 68 | 73 | 66 | 59 | 51 | 68 | 82 | 51 | 71  | 63  | 54  | 88  | 66  | 86  | 67.60 |

---

### Official Review · Reviewer_Tp5K · 2025-11-01

**Soundness:** 3
**Presentation:** 3
**Contribution:** 3
**Rating:** 6
**Confidence:** 2

**Summary:**

This paper introduces Lifelong Embodied Navigation Learning, a new problem setting where an agent must continually learn a sequence of navigation tasks across diverse scenes and instruction styles without suffering from catastrophic forgetting. To address this, the authors propose Uni-Walker, a novel framework built upon large language models. The core of Uni-Walker is a Decoder Extension LoRA architecture that decouples knowledge into a task-shared subspace and multiple task-specific "expert" subspaces. The framework includes specific strategies to manage this knowledge: a KIS and anECAS to learn shared knowledge, and an ESOC and NSCoT to learn distinct, task-specific skills. At inference time, a TAKA mechanism selects the most relevant experts for the current task. The authors create a new benchmark for LENL and demonstrate through extensive experiments that Uni-Walker significantly outperforms existing continual learning methods, achieving a higher success rate while drastically reducing forgetting.

**Strengths:**

- Novel Problem: The paper formally introduces and tackles the problem of lifelong learning for embodied navigation, a crucial challenge for developing general-purpose agents.
- Comprehensiv Method: Uni-Walker is a well-thought-out framework that addresses the core challenges of knowledge transfer and catastrophic forgetting by explicitly decoupling and managing shared and task-specific knowledge.
- Rigorou Evaluation: The experimental validation is a major strength. The creation of a new benchmark, comparison against a wide array of strong baselines, and thorough ablation studies for each component of the model provide convincing evidence for the method's effectiveness.
- Significant Performance Gains: The results are not incremental. Uni-Walker shows large improvements in both task success and resistance to forgetting.

**Weaknesses:**

System Complexity: The final Uni-Walker model is a composite of many components (KIS, ECAS, SSC, ESOC, NSCoT, TAKA). While the ablation studies justify each part's inclusion, the overall system is quite complex. A discussion on the relative importance of these components or potential simplifications would be beneficial.

**Questions:**

Could you comment on the scalability of the Uni-Walker framework as the number of sequential tasks grows into the hundreds? Specifically, how does the performance of the Task-Aware Knowledge Aggregation (TAKA) mechanism degrade, if at all, as the pool of retrievable experts becomes very large?

---

> ### Author Response · Authors · 2025-11-20
> **Response to Reviewer Tp5K (1 / 2)**
>
> Thank you for the careful reading and constructive comments. We are grateful for your recognition of the proposed Lifelong Embodied Navigation Learning task and the strength of our Uni-Walker. Below, we respond to all the concerns.
>
> **1. A discussion on the relative importance of KIS, ECAS, SSC, ESOC, NSCoT, and TAKA components or potential simplifications would be beneficial. (Weaknesses-1)**
>
> Thank you for the thoughtful suggestion. As shown in our ablation studies (Tables 4–6), every component of Uni-Walker provides performance improvement. Below, we summarize their roles and contributions.
>
> **Knowledge Inheritance Strategy (KIS).** KIS equips Uni-Walker with a human-inspired capability to efficiently acquire new skills by leveraging previously learned shared knowledge. It initializes the new expert subspace using relevant past knowledge (Eq. 2), enabling efficient adaptation. Removing KIS results in a 7.0% SR degradation.
>
> **Experts Co-Activation Strategy (ECAS).** ECAS is the core architectural mechanism of DE-LoRA: it directly activates previously learned experts that are most relevant to the current task (Eq. 3), enabling knowledge sharing across tasks. Without ECAS, SR decreases by 9.2%.
>
> **Shared Smoothing Consolidation (SSC).** SSC is to progressively refine the shared subspace $A$, mitigating catastrophic forgetting of previous tasks (Eq. 4). Removing SSC leads to a 7.6% SR degradation.
>
> **Expert Subspace Orthogonality Constraint (ESOC).** ESOC performs orthogonal constraint across expert subspaces $B$, promoting decoupled representation of navigation knowledge and preventing subspace mixing (Eq. 7). Its removal causes 3.8% SR degradation.
>
> **Navigation-Specific Chain of Thought (NSCoT).** NSCoT provides task-style–specific reasoning for different instruction styles. It invokes relevant pretrained LLM knowledge and guides the agent toward more targeted decision-reasoning. Removing NSCoT yields the largest drop (16.2% SR), showing that style-aware reasoning is crucial for LLM-based navigation
>
> **Task-Aware Knowledge Aggregation (TAKA).** TAKA is indispensable for task-ID–agnostic inference. It selects the most relevant experts using mixed matching and enables Uni-Walker to perform navigation in an ID-agnostic task. We also provide ablation (Table 6) that validates the effectiveness of the mixed matching strategy used by TAKA.
>
> In summary, ECAS, TAKA, and NSCoT are the core components for accomplishing the LENL task, while the training strategies KIS, SSC, and ESOC promote disentangled representation of embodied navigation knowledge and further improve performance. We suggest that since ESOC has a relatively low impact (only +3.8%), it can be selectively removed when users have to simplify the Uni-Walker architecture.

---

> ### Author Response · Authors · 2025-11-20
> **Response to Reviewer Tp5K (2 / 2)**
>
> **2. The scalability of the TAKA as the pool of retrievable experts becomes very large. (Questions-1)**
>
> Thank you for the insightful comment. The TAKA maintains good scalability even when the number of tasks grows to a very large scale. TAKA performs embedding-level retrieval using the large-scale pretrained CLIP model, and activates only the Top-K experts based on embedding similarity (Eq. 10). Since the matching process relies on **relative similarity rather than absolute similarity scores**, the retrieval quality does not degrade even as the total number of experts increases. i) To validate this, we conduct an extended retrieval experiment across all 90 scenes in the Matterport3D Simulator. For each scene, the agent randomly navigates 100 times and performs TAKA matching. The results show that the retrieval success rate remains consistently high and stable regardless of the number of tasks:
>
> | Metric  | 10 Tasks | 20 Tasks | 30 Tasks | 40 Tasks | 50 Tasks | 60 Tasks | 70 Tasks | 80 Tasks | 90 Tasks |
> |:---:|:--------:|:--------:|:--------:|:--------:|:--------:|:--------:|:--------:|:--------:|:--------:|
> | Retrieval SR |  99.8%   |  99.7%   |  99.8%   |  99.7%   |  99.8%   |  99.7%   |  99.8%   |  99.7%   |  99.8%   |
>
> These results demonstrate that TAKA's matching robustness does not diminish as the number of experts grows, confirming its practical scalability. ii) To further evaluate scalability under extremely large task pools, we conduct an additional scalability test. Specifically, after training on the first 10 tasks, we artificially add 1000 randomly initialized experts with their corresponding 1000 retrieval embeddings extracted from random natural scenarios by CLIP, simulating a very large-scale lifelong learning scenario. We then continue training Uni-Walker on the next 5 tasks and measure performance on tasks S10–S15. The SR (%) results are summarized below:
>
> | Methods    | S10 | S11 | S12 | S13 | S14 | S15 |  Avg. SR |
> |-------------------|-----|-----|-----|-----|-----|-----|----|
> | Uni-Walker        | 73  | 62  | 53  | 88  | 65  | 86  |  71.2  |
> | 1000+ Uni-Walker  | 74  | 61  | 54  | 88  | 64  | 86  |  71.2  |
>
> The results show no degradation when Uni-Walker operates alongside 1000 additional experts. This confirms that both training stability and TAKA retrieval remain robust under very large expert pools. In addition, based on the theoretical experience provided in [1], with the CLIP latent space dimension of 1024, the maximum number of retrievable results is 4,031,487 (1024^3 × 0.0037 + 1024^2 × 0.0520 + 1024 × 4.0309 − 10.5322).
>
> iii) In addition, the storage cost of TAKA is lightweight. Each navigation task requires only 150 KB for visual embeddings and 3.1 KB for instruction embeddings. Thus, even with 1,000+ tasks, the total storage cost is still modest: 1000 × 154 KB ≈ 150.39 M. Such overhead is negligible for LLM systems (7B, 13B, or larger), further supporting the scalability of TAKA in large-scale lifelong learning scenarios.
>
> [1] On the Theoretical Limitations of Embedding-Based Retrieval. arXiv 2025.

---

### Author Response · Authors · 2025-11-20
**Response to All**

Dear reviewers and area chairs:

We extend our gratitude to all the reviewers and area chairs for dedicating their time and effort to evaluating our paper. We also thank the reviewers for their positive and insightful comments, which can help us improve our work.

We are encouraged that:

* All the reviewers (Tp5K, 5sW8, AqL1, and Fhsy) agree that our work explores an **unexplored but practically important task**, which requires agents to continually learn a sequence of navigation tasks across diverse scenes and instruction styles for developing general-purpose agents.

* All the reviewers recognize that we **establish a new benchmark** for the proposed LENL task with varying scenes and instruction styles.

* Reviewer Tp5K, 5sW8, and Fhsy think that our Uni-Walker design provides **novel framework** or **high originality** for the specific challenges of LENL.

* Reviewer AqL1 thinks that our research **would possibly bring new research works** for navigation.

* All reviewers recognize that our model achieves **state-of-the-art performance under comprehensive experiments**.

We appreciate the opportunity to discuss and refine our Uni-Walker. We have responded to all reviewers individually to address the concerns, and the following is a brief summary:

* For Reviewer Tp5K and 5sW8, we provide a detailed analysis incorporating ablation experiments to clarify the contribution of each component in Uni-Walker.

* For Reviewer Tp5K and 5sW8, we add experiments to validate the scalability and robustness of Uni-Walker under thousands of tasks.

* For Reviewer 5sW8 and Fhsy, we add deployment experiments to validate the performance of Uni-Walker in real-world environments.

* For Reviewer 5sW8, we add sensitivity analysis of the most critical hyperparameters.

* For Reviewer 5sW8, we clarify how the appropriate NSCoT is selected during inference.

* For Reviewer 5sW8, we clarify how the same instruction knowledge is identified in the KIS.

* For Reviewer AqL1, we further clarify the key novel design in the proposed framework.

* For Reviewer Fhsy, we add comparisons with non-LoRA continual-learning baselines.

* For Reviewer Fhsy, we discuss the boundaries of our LENL task.

We have highlighted all modifications in the revised paper in blue. We hope these additions address the reviewers’ concerns and further improve our work. If any further clarifications or suggestions would help strengthen the paper, we would be happy to address them and incorporate the changes into the final version. Thank you again for your time and efforts!

Best,

Authors of Paper #44

---

> ### Author Response · Authors · 2025-12-02
> **Summary of Reviewer Discussions**
>
> Dear area chairs and reviewers,
>
> We sincerely thank the area chairs for their time and efforts in managing our submission. We are also grateful to all reviewers for their careful reading and constructive comments. Below we briefly summarize how the reviewers' opinions evolved after our rebuttal and discussion.
>
> **Reviewer Fhsy**
> Reviewer Fhsy initially raised some experimental suggestions and applicability discussions. After our rebuttal, Reviewer Fhsy stated that he/she **had read other reviews and responses and found no additional concerns**. The reviewer decided to maintain the original positive rating.
>
> **Reviewer AqL1**
> Reviewer AqL1 initially raised points regarding methodological contributions and suggested revisions to several figures. After our rebuttal, Reviewer AqL1 stated that **the main concern is addressed**. The reviewer kept the original positive rating.
>
>
> Due to unforeseen circumstances, Reviewer 5sW8 and Reviewer Tp5K were unable to respond. However, in their initial reviews, **both reviewers already assigned a score of 6**, which demonstrates recognition of this work. We have responded to the reviewers’ comments point by point to address all concerns.
>
>
> **Reviewer 5sW8**
>
> Reviewer 5sW8 raised four main concerns:
> - **Component contributions & complexity–performance trade-offs**: We provided a detailed analysis incorporating ablation experiments to clarify the contribution of each component in Uni-Walker and discussed the trade-offs between model complexity and empirical performance.
> - **Computational scalability with number of tasks**: We added experiments to validate the scalability and robustness of Uni-Walker under thousands of tasks.
> - **Sensitivity analysis of critical hyperparameters**: We added sensitivity analyses for the most critical hyperparameters.
> - **Algorithmic implementation details**: We clarified how the appropriate NSCoT is selected during inference and how the same instruction knowledge is identified in the KIS.
>
> **Reviewer Tp5K**
>
> Reviewer Tp5K raised two main concerns:
>
> - **Component contributions & potential simplifications**: We provided a detailed analysis incorporating ablation experiments to clarify the contribution of each component in Uni-Walker and discussed potential simplifications of the pipeline.
> - **Scalability with increasing tasks**: We added experiments to validate the scalability and robustness of Uni-Walker under thousands of tasks.
>
> We hope this summary, together with the rebuttal and discussion, clearly reflects how reviewers' concerns were addressed. We sincerely thank the area chairs again for their time and efforts in managing this process.
>
> Best regards,
>
> Authors of Paper #44

---

### Meta-Review · Area_Chair_ec4g · 2026-01-04

**Summary:**

The paper formalizes lifelong embodied navigation learning (LENL), where an agent sequentially adapts to navigation tasks across new scenes and diverse instruction styles while retaining prior navigation knowledge to mitigate catastrophic forgetting, and introduces a dedicated benchmark for training and evaluation.
It proposes uni-walker, a continual navigation framework that decouples task-shared and task-specific knowledge via decoder extension lora (DE LoRA), enabling more effective learning of new tasks while previously acquired capabilities are preserved.
For shared knowledge, knowldge inheritance and experts co-activation are designed to transfer and continually refine common navigation knowledge across tasks. For specific knowledge, an expert subspace orthogonality constraint plus a navigstion-specific chain-of-thought mechanism capture task-specific representations and tailor reasoning to instruction styles (VLN, OLN, DUN).
Extensive experiments are reported to show uni-walker learns new navigation tasks without forgetting and generalizes to unseen tasks, delivering strong performance in the proposed LENL setting.

The committee appreciated the formalization of LENL and the accompanying benchmark (reviews by: tp5k; 5sw8; aql1; fhsy). The committee also highlighted the uni-walker design that decouples task-shared and task-specific knowledge via DE-LoRA with expert routing and aggregation (R# tp5k; 5sw8; fhsy). The rigorous empirical validation was recognized as well, including broad baseline comparisons and component-wise ablations that show sizable gains in success and SPL with markedly reduced forgetting (reviewers tp5k; 5sw8; fhsy). The overall clarity and the non-incremental nature of the reported improvemnts were also consistently noted across reviews, even as some assessments were framed more conservatively.

The rebuttal clarified novelty & component roles with quantitative ablations, and addressed scalability and generalization questions through added retrieval-at-scale tests, hyperparameter sensitivity, non- LoRA baselines, and preliminary real-world deployments. Reviewers indicated that the key concerns were largely addressed and kept their overall assessments unchanged (reviewers: aql1; fhsy). These additions and the figure readability fixes should be incorporated into the camera-ready revision. The area chair concurs with the strong convergence in the committee reviews and post-rebuttal discussion in favor of acceptance.

**Reviewer Concerns:**

Most concerns were addressed in the rebuttal with added evidence. Component importance and simplification were clarified via quantitative ablations and an explicit “core vs optional” breakdown (R#tp5k; R#5sw8). Scalability concerns were addressed with retrieval-at-scale tests for TAKA plus storage and compute accounting for per-task experts and related statistics (tp5k; 5sw8). Additional items were covered via hyperparameter sensitivity, a dedicated unseen-generalization table, and clarifications on inference-time instruction-style handling and expert grouping, including an automatic variant (R#5sw8). Breadth concerns were partially mitigated with preliminary real-world deployments and added comparisons to selected non-LoRA continual learning baselines, plus clarification of task boundaries and an automatic boundary variant (R-fhsy; R-5sw8). Figure readability issues were revised (R-aql1). Remaining limitations are mainly scope: primary evaluation still Matterport3D; larger and more diverse lifelong sequences and further real-world validation remain future work.

**Reviewer Scores:**

* tp5k: Main questions (component roles; TAKA scalability) addressed with new ablations and scalability experiments; no post-rebuttal update, but likely same score, possibly higher.
* 5sw8: Main questions (scaling; sensitivity; generalization breakdown; inference clarifications; added baselines and real-world) addressed; no post-rebuttal update, but likely same score, possibly higher.
* aql1: Stated main concern addressed; kept rating.
* fhsy: Stated no additional concerns post-rebuttal; kept rating.

---

### Decision · Program_Chairs · 2026-01-26

Accept (Poster)